# Segmentation-Based Multi-Pixel Cloud Optical Thickness Retrieval Using a Convolutional Neural Network

Vikas Nataraja[1], K. Sebastian Schmidt[1, 2], Hong Chen[1, 2], Takanobu Yamaguchi[3, 4], Jan Kazil[3, 4], Graham Feingold[3], Kevin Wolf[1], and Hironobu Iwabuchi[5]

[1]Laboratory for Atmospheric and Space Physics (LASP), University of Colorado, Boulder, CO 80303, USA
[2]Department of Atmospheric and Oceanic Sciences, University of Colorado, Boulder, CO 80303, USA
[3]Cooperative Institute for Research in Environmental Sciences (CIRES), University of Colorado Boulder, CO 80309, USA
[4]National Oceanic and Atmospheric Administration (NOAA), Chemical Sciences Laboratory, Boulder, CO 80305, USA
[5]Center for Atmospheric and Oceanic Studies, Graduate School of Science, Tohoku University, Sendai, Miyagi 980-8578, Japan

**Correspondence:** Vikas Nataraja (Vikas.HanasogeNataraja@lasp.colorado.edu)

**Abstract.**

We introduce a new machine learning approach to retrieve cloud optical thickness (COT) fields from visible passive imagery. In contrast to the heritage Independent Pixel Approximation (IPA), our Convolutional Neural Network (CNN) retrieval takes the spatial context of a pixel into account, and thereby reduces artifacts arising from net horizontal photon transfer, commonly known as independent pixel (IP) bias. The CNN maps radiance fields acquired by imaging radiometers at a single wavelength channel to COT fields. It is trained with a low-complexity and therefore fast U-Net architecture where the mapping is implemented as a segmentation problem with 36 COT classes. As a training data set, we use a single radiance channel (600 nm) generated from a 3D radiative transfer model using Large Eddy Simulations (LES) from the Sulu Sea. We study the CNN model under various conditions based on different permutations of cloud aspect ratio and morphology, and use appropriate cloud morphology metrics to measure the performance of the retrievals. Additionally, we test the general applicability of the CNN on a new geographic location with LES data from the equatorial Atlantic. Results indicate that the CNN is broadly successful in overcoming the IP bias and outperforms IPA retrievals across all morphologies. Over the Atlantic, the CNN tends to overestimate the COT but shows promise in regions with high cloud fractions and high optical thicknesses, despite being outside the general training envelope. This work is intended to be used as a baseline for future implementations of the CNN that can enable generalization to different regions, scales, wavelengths, and sun-sensor geometries with limited training.

## 1 Introduction

Cloud optical properties play an important role in determining the cloud radiative effect (CRE), surface energy budget, heating profiles, etc. Cloud optical thickness (COT) is important for the shortwave CRE. Accurately predicting the COT will help to improve our understanding of the energy budget. Currently, the most-used cloud optical properties are retrieved under the independent pixel approximation (Vardhan et al., 1994), or IPA, which assumes clouds are homogeneous within the pixel and is blind to the spatial context of adjacent pixels.

## 1.1 Effects of Cloud Inhomogeneity

In the real world, clouds are inhomogeneous. Cloud spatial inhomogeneity effects on atmospheric radiation and remote sensing have been studied extensively for decades. To appreciate that, one only needs to consider that the Stephens and Tsay (1990) review paper on the once prominent cloud absorption anomaly was itself the synthesis of a body of work starting in the 1960s. This anomaly is understood as the discrepancy between the absorption as calculated from in-situ cloud microphysics measurements and as inferred from measured shortwave net irradiances above and below a cloud layer. Rawlins (1989) and others identified net horizontal photon transport $H$ as the potential cause. This term is an important addition to the energy conservation of a layer with finite horizontal extent (domain size),

$$R + T + A = 1 + H \tag{1}$$

where $R$, $T$, and $A$ are the irradiances that are reflected, transmitted, and absorbed by the layer, normalized by the incident irradiance (Marshak and Davis, 2005), whereas $H$ quantifies the net lateral exchange between the domain captured in the equation above and its surroundings. It can only be neglected if clouds are horizontally homogeneous over a sufficiently large domain. In practice, this condition is not met very often, and yet one-dimensional radiative transfer (1D-RT) makes precisely that assumption. The $H$ term can thus be regarded as the missing physics in 1D-RT, which largely explains the lack of radiative closure between measured and calculated $A$, $T$, and $R$ in earlier studies (Marshak et al., 1999; Kassianov and Kogan, 2002; Schmidt et al., 2010; Kindel et al., 2011; Ham et al., 2014; Song et al., 2016).

Barker and Liu (1995), hereafter referred to as BL95, first quantified the effect of horizontal photon transport on COT retrievals with Landsat data. Interpreting their Landsat-derived COT fields as truth, they calculated synthetic radiance fields with a Monte-Carlo 3D-RT model and subsequently retrieved COT from those, emulating the IPA retrieval process with realistic clouds. They found that the optical thickness of optically thick clouds is underestimated, whereas optically thin clouds appear thicker than they really are. Because of horizontal photon transport, the "dark" pixels collectively brighten at the expense of the "bright" pixels. The magnitude of such errors, quantified by retrieval performance metrics introduced in Sect. 3.4, depends on cloud type and morphology (horizontal distribution, geometric thickness and other parameters).

To some degree, radiance averaging in spatially coarse pixels decreases the independent pixel (IP) bias because net horizontal photon transport drops off with larger pixels. On the other hand, radiance averaging also leads to the so-called plane-parallel (PP) bias because the reflectance $r$ is a concave function of COT, and therefore $r(\langle COT \rangle) \geq \langle r(COT) \rangle$. In other words, reflectance as a function of the mean of the optical thickness is always greater than or equal to the mean of the reflectance of the optical thickness. The PP bias *increases* with pixel size, while the IP bias *decreases*. To reduce the PP bias, Cahalan (1994) introduced the concept of effective COT for marine stratocumulus clouds, which can be parameterized as a function of $\langle COT \rangle$ and the standard deviation of the logarithm of the COT. For stratocumulus and some other boundary layer clouds, the optimum (i.e., the minimum of IP and PP combined) occurs at a scale of about 1 km (Davis et al., 1997; Zinner and Mayer, 2006), which

is why currently operational cloud retrievals are performed at this scale (e.g. Platnick et al., 2021).

In addition to net horizontal photon transport, there are other mechanisms causing inhomogeneity biases in cloud retrievals, most notably shadowing, which is especially significant for low sun elevation and pronounced cloud top variability, which leads to roughening of the retrieved COT fields (Marshak et al., 2006; Iwabuchi, 2007) – as opposed to smoothing that is

caused by horizontal photon transport. The retrieval of droplet effective radius (REF), which is retrieved along with COT in the bi-spectral technique by Nakajima and King (1990), is also affected by cloud inhomogeneity biases (Marshak et al., 2006; Zhang et al., 2012), as are downstream parameters such as the liquid water path (LWP) and the cloud droplet number concentration. The occurrence of smoothing and roughening as manifested in power spectra and autocorrelation functions varies by cloud type and scale, imager wavelength, as well as solar zenith angle (Oreopoulos et al., 2000). Iwabuchi and Hayasaka

(2002) distinguished geometric inhomogeneity (morphology, thickness and cloud top roughness), horizontal variance, resolution and scaling (power spectrum exponent), and sun-sensor geometry as the primary drivers of biases and retrieval noise of IPA retrievals. Of these, Várnai and Davies (1999) specifically compared cloud-top and horizontal variability with the tilted IPA (TIPA) and found that COT variations caused by the variability of geometric thickness rather than by the extinction coefficient lead to greater reflectance biases, at least for oblique geometries. They also quantified different sub-mechanisms such as

upward and downward "trapping" and "escape" of photons, and proposed to treat them separately in future correction schemes.

Figure 1 illustrates the magnitude of the problem for cumulus clouds. Synthetic radiances obtained from an LES cloud scene with a 3D-RT model (Sect. 2.2) emulate imagery observations. From those radiances, COT fields were retrieved via IPA. For the clouds shown as an inset in Fig. 1a as an example, the IPA retrieval (Fig. 1c) significantly underestimates the ground truth

COT (Fig. 1b) due to $H$ and other 3D effects.

## 1.2   Statistical Mitigation of Cloud Inhomogeneity Effects

Tremendous effort has been made to mitigate the effects of cloud inhomogeneity. Early mitigation efforts employed statistical approaches. For example, BL95 determined the slope $\delta$ for the logarithmic relationship between the IPA-retrieved COT $\tau_{IPA}$

and the true COT $\tau_{true}$ as:

$$\tau_{IPA} = \tau_{true}^{\delta} \qquad (2)$$

where $\delta$ is parameterized as a function of the cloud geometric thickness $h$ for the specific cloud fields used:

$$\delta = e^{-h/h_0} \text{ for } h < 1{,}000 \text{ m} \qquad (3)$$

with $h_0 \approx 2km$. The corrected COT can then be obtained as $\tau_{IPA}^{1/\delta}$. Chambers et al. (1997) tabulated aspect-ratio dependent

corrections in the form of

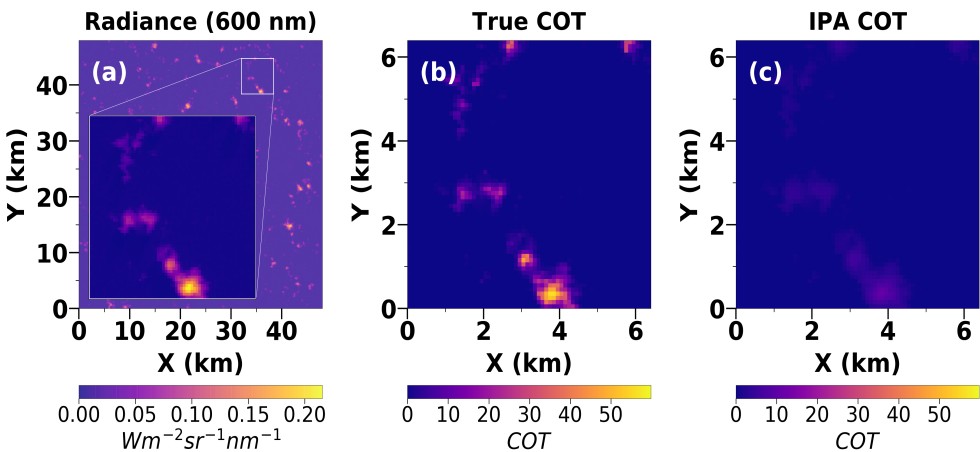

**Figure 1. (a)** Synthetic radiance field at 600 nm generated with EaR³T using LES for a nadir view (30° solar zenith angle, 0° solar azimuth angle) that would be seen by a satellite imager at 705 km. Inset shown is a 6.4 km x 6.4 km sub-domain region. **(b)** True COT (ground truth) for the sub-domain of 6.4 km x 6.4 km. **(c)** IPA retrieval for the sub-domain. The underestimation made by the IPA retrieval is visually clear.

$$\tau_{IPA} = \tau_{true}(1 - o - bv) \tag{4}$$

where $o = 0.01 \ldots 0.05$ and $b = 0.15 \ldots 0.25$ are SZA-dependent fit parameters, and the formula is inverted to un-bias the COT. In this case, the aspect ratio $v$ is derived from satellite observations as the ratio of the cloud top variability to the horizontal e-folding distance of the COT auto-correlation function. It is really a metric of cloud top roughness, but serves as a proxy for the true aspect ratio. In contrast to the BL95 formula, the Chambers et al. (1997) parameterization only corrects for the underestimation of COT for large values, not for the overestimation at small COT.

Iwabuchi and Hayasaka (2002) introduced more complex statistical parameterizations that account for sun-sensor geometry and cloud morphology among other factors, with the main objective of correcting the first two moments of the COT probability distribution function (PDF) (mean value and variance, Iwabuchi, 2007). Marshak et al. (1998) developed a non-local IPA (NIPA) that considers pixel-to-pixel interactions by adding a convolution kernel to the IPA that reproduces the observed Landsat scale break, and inverted the approach for the recovery of the true COT power spectrum from observed radiance fields. To stabilize (de-noise or smooth) this deconvolution process, they used spatial regularization. Zinner et al. (2006) applied a similar approach to aircraft radiance observations of broken clouds, but implemented it as a step-wise sharpening algorithm, which adjusts the point-spread function of the deconvolution kernel iteratively until the calculated radiance fields match the observations. Applied to synthetic observations, it not only re-creates the original power spectrum of the underlying LWP field, but also reproduces the original PDF.

## 1.3 Cloud Inhomogeneity Mitigation Using Tomography and Neural Networks

Another promising mitigation strategy for 3D cloud biases is tomography (e.g. Forster et al., 2021) where multi-angle radiance observations are inverted to retrieve not only cloud boundaries (through stereo reconstruction or space carving), but also the 3D distribution of parameters such as the liquid water content (LWC) and REF. This is done by iteratively adjusting the inputs to 3D-RT calculations until the output is consistent with the observations – an approach that has recently become tractable (Levis et al., 2020). Tomography does not require training and comes with built-in closure between the observed and calculated radiance fields. However, it requires multi-angle radiances and extensive RT calculations, which are computationally expensive.

Pixel context-aware algorithms have become a promising approach for resolving cloud inhomogeneity effects when retrieving cloud optical properties from radiance measurements. Faure et al. (2002) implemented a mapping neural network (MNN) where the solution to the inverse problem is understood as mapping from radiance to COT not only on the individual pixel basis (as in IPA), but also from neighboring pixels. The transfer functions from neighboring pixels are coefficients that are learned iteratively by the MNN with training data. They can be understood as spatial filters. This is similar to the idea of an averaging kernel from Marshak et al. (1998), but more general and applied in the opposite direction (from radiance fields to COT). Cornet et al. (2004) applied this approach for the estimation of domain-averaged COT and REF. Iwabuchi (2007) built on the idea of spatial mapping, but generalized it further to include other wavelengths. The filter coefficients are determined by regression using least-square fitting based on synthetic training data. Instead of mapping directly to COT space, the observed radiance fields are mapped to pseudo-IPA radiance fields where 3D effects are removed, and from where the standard IPA technique can be applied to infer COT and other parameters.

Such pattern-driven image analysis proliferated with the advent of deep learning, specifically Convolutional Neural Networks (CNNs, LeCun et al., 1989, LeCun et al., 1998). In the last two decades, the increased use of GPUs (Graphics Processing Units) has enabled the efficient processing of large training data sets for increasingly complex networks (Krizhevsky et al., 2012). Therefore, it was only natural to harness these techniques for 3D cloud remote sensing. Okamura et al. (2017) presented the first CNN for multi-pixel cloud reflectance retrievals for COT and REF that was trained with LES-based synthetic data, followed by Masuda et al. (2019) who developed a CNN retrieval of the slant COT from ground-based observations by a radiometrically calibrated fish-eye lens camera (see Sect. 2). Sde-Chen et al. (2021) combined the two worlds of CNNs and tomography to reconstruct the 3D cloud extinction field using multi-view satellite images. These algorithms demonstrated that a CNN is capable of recovering the original cloud fields with higher fidelity than previous techniques, albeit only after a significant training effort sometimes involving days of training using supercomputers.

Since the magnitude of 3D remote sensing biases depends on cloud spatial characteristics, CNNs have the potential to outperform their regression-based predecessors (Sect. 1.2). In this work, we introduce a CNN that builds on previous work (Masuda et al., 2019), but significantly reduces training time through:

1. reduced complexity of the architecture (Sect. 3.2);

2. a deliberately minimal training data set that is still general enough to make the trained CNN applicable to a wide range of conditions, while outperforming the IPA in terms of the retrieval performance metrics we introduce in Sect. 3.4.

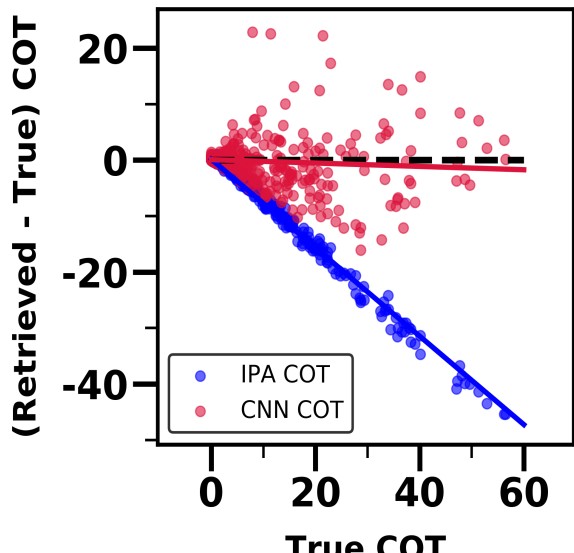

**Figure 2.** A scatter plot comparing the COT retrieved by the IPA method (blue scatter) and the CNN (red scatter) as a function of the true COT. The dashed black line depicts the ideal retrieval having a slope of 0. The solid blue and red lines are linear regression lines fitted to the IPA and CNN retrievals respectively.

For the IPA-retrieved COT from Fig. 1c, we show the retrieval bias in Fig. 2, which is the difference between the retrieved COT and the ground truth COT, and expressed as a function of the ground truth COT. Figure 2 also provides a preview of the results of the CNN that we describe in more detail later. The dependence of the bias on the ground truth can be parameterized through linear regression. In this case, the slope of the regression for the IPA retrieval is -0.79, which reflects the significant underestimation of the COT for the majority of the pixels from Fig. 1c. In contrast, the CNN retrieval shows a much smaller bias with a slope of -0.03, though the scatter is not reduced relative to the IPA. More details on the definition of the various retrieval performance parameters such as the slope and scatter are described in Sect. 3.4. It is worth noting that the IPA retrieval in Fig. 2 does appear to have a linear relationship with the difference between the retrieved and true COT which would imply that it is indeed possible to parameterize this effect as a 3D correction. Furthermore, as we discuss in Sect. 1.2, there have been approaches that have attempted to do so, including Iwabuchi and Hayasaka (2002). However, the underlying problem with such a method is that the parameters are fixed, and derived for very specific cloud fields using multivariate fitting. By contrast, with our proposed CNN (and its future iterations), the intention is to utilize the existing spatial context in cloud imagery to learn the

underlying features that can then be generalized and applied to correct 3D radiative/net horizontal photon transport effects.

In our paper, we use two LES data sets from distinct regions of the globe as 3D-RT input (Sect. 2.2) to generate synthetic radiance data that a satellite would observe at nadir. We then train the model with 6.4 km x 6.4 km resolution radiance images as the input and COT as the truth from the first LES data set (shallow convection near the Philippines). The LES data set we chose contains six distinct cloud morphologies that correspond to a locally representative range of aerosol and wind shear conditions. We validate the performance of the CNN with unseen image pairs from the first data set by assessing a number of retrieval performance parameters as a function of cloud field parameters such as mean COT and cloud fraction (CF). Along the way, we test different training data selection criteria that increase the capacity of the trained network to generalize. Finally, we test the CNN trained on data from the Philippines on the second data set (closed and open-cell shallow convection in the Southeast Atlantic) to gauge its functional capacity under completely different circumstances.

Section 2 describes the generation of the training and validation data from the LES data, including the 3D-RT, followed by the CNN architecture and methodology in Sect. 3. Section 4 discusses the evaluation of the CNN under various experiments and case-studies in data conditioning and selection. In Sect. 5, we discuss the main takeaway points from our study and outline a path for the future.

## 2 Data Generation

To train the CNN, we need input and true data. The input data consists of radiance images, each at 600 nm wavelength. For each input image, a ground truth image of the same size is used. In our case, the COT is used as the ground truth. This true image goes through a series of processing steps explained in Sect. 2.3. To generate this data, we use two LES models in two regions, coupled with radiance calculations (Sect. 2.2).

### 2.1 Large Eddy Simulations

The two LES data sets were chosen from regions where NASA aircraft field campaigns were conducted in the recent past to allow for direct validation of CNN-based retrievals with in-situ microphysics and radiation measurements in future studies. The first LES data set (Sect. 2.1.1) is based on 7SEAS (Southeast Asian Studies) ship-based observations in the Sulu Sea area, which was also sampled during the 2019 NASA CAMP$^2$Ex (Cloud, Aerosol and Monsoon Processes-Philippines Experiment) aircraft campaign. The second data set (Sect. 2.1.2) was based on the 2017 Cloud-Aerosol-Radiation Interactions and Forcing (CLARIFY) campaign in synergy with the 2018 campaign of the NASA airborne ORACLES (ObseRvations of Aerosols above CLouds and their intEractionS) study (Redemann et al., 2021) that took place in the Southeast Atlantic. Both data sets are dominated by shallow convection, but with different attributes.

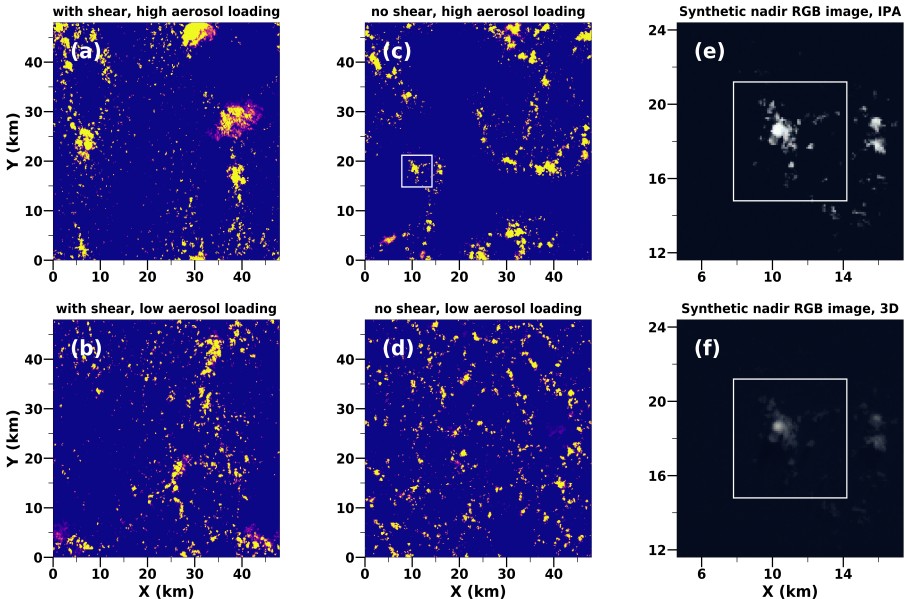

**Figure 3. a-d**: LES COT fields from the 7SEAS campaign in the Sulu Sea with/without wind shear and with low/high aerosol loading (the maximum COT is capped at 4 to emphasize low-COT regions); **e, f**: Synthetic radiance calculations (Red = 600 nm, Green = 500 nm, Blue = 400 nm) with IPA and 3D, shown for the 6.4 km x 6.4 km sub-domain (white box) in **(c)**. All observations are of a hypothetical satellite imager at an altitude of 705 km with a solar zenith angle of 30° (with nadir viewing angle) and a solar azimuth angle of 0°.

For the Sulu Sea data set, Fig. 3 shows how wind shear and aerosol loading affect cloud morphology. For conditions with no wind shear, one can see a high level of organization (hexagonal walls of convection), especially for the low aerosol loading case shown in Fig. 3b and Fig. 3d. We selected six of these open-cell convection cases and sampled 64 x 64 pixel sub-domains spanning 6.4 km x 6.4 km from the COT fields and the corresponding radiance fields as shown by the white box in Fig. 3c. Figures 3e and 3f visualize the difference in the radiance level between IPA and 3D radiance calculations respectively for the highlighted sub-domain. Figure 3a shows a scenario with vertical wind shear and high aerosol loading. We used the native horizontal resolution of the simulations (100 m) as the pixel size for the synthetic radiance simulations – a scale where the IPA bias dominates over the PP bias and can therefore be optimally studied here.

The six 48 km x 48 km scenes each generated a hundred 6.4 km x 6.4 km training image pairs. They were obtained by clipping off 64-pixel stripes on all sides to avoid edge effects from the cyclic boundary conditions in the LES and 3D-RT, and subsequently moving a 64 x 64 pixel selector window across the remaining domain with a horizontal and vertical stride of 32 pixels. For CNN training, the original number of samples is very low. Therefore, we augmented the native-resolution training pairs by horizontally coarsening the fields by a factor of 2, such that each original 100 m x 100 m cell was assigned a spatial extent of 400 m x 400 m and then split into four cells, leaving the vertical resolution of the fields (40 m) intact. In addition to providing additional training pairs after sub-sampling as described for the native-resolution data, this coarsening procedure

also effectively generates horizontally smoother cloud fields while halving the cloud aspect ratio (cloud height divided by cloud width) since we only change the horizontal resolution. In other words, one of the key drivers for 3D COT biases as described by BL95 and others is systematically changed in the training data to introduce some training data diversity. A subsequent second coarsening step introduces another level of coarsening and the aspect ratio has now reduced by a factor of 4 from the original. The three data sets, labeled 1 x 1 (native resolution), 2 x 2, and 4 x 4, respectively, are used separately (Sect. 4.1) to examine the impact of the cloud aspect ratio on the retrieval performance, and together (Sect. 4.2) to assess the impact of training sample number along with algorithm robustness and accuracy for a physically more diverse data set. A more consolidated version of the three data sets is evaluated to decrease training time (Sect. 4.3).

The Atlantic data set (Sect. 2.1.2) encompasses both open-cell and closed-cell convection, from which we sampled five 350 x 350 pixel scenes. These data were only used at the native resolution (100 m x 100 m x 40 m voxel size as the Sulu Sea simulations); no data augmentation was necessary because the fields only served as validation. This is further explained in Sect. 4.4.

### 2.1.1 Sulu Sea Data Set

The simulation configurations were designed to investigate aerosol-cloud interactions in trade cumulus cloud fields in the Philippine areas as a pilot study for the CAMP[2]Ex field program (Reid et al., 2022). The detailed model configurations and scientific findings were reported by Yamaguchi et al. (2019). The initial, environmental, and boundary conditions were based on a ship measurement that took place on September 21 in the Sulu Sea during the 7 Southeast Asian Studies (7SEAS) campaign in 2012 (Reid et al., 2016). In addition to 6 hourly data, the hourly ERA5 (Hersbach et al., 2020) data were supplementary. A total of 6 simulations were performed with / without vertical wind shear with three different aerosol number concentrations - 35, 150, and 230 mg$^{-1}$ - for 60 hours with a 48 km x 48 km domain and two-moment bin microphysics scheme. These simulations revealed that trade cumulus clouds organize so that they produce a similar amount of precipitation and cloud radiative effect, which is consistent with a buffering of the aerosol effect as discussed by Stevens and Feingold (2009). Vertical wind shear was found to impose two effects which compensate one another; the wind shear enhances clustering of clouds, which tends to protect clouds from being evaporated, while it tilts the clouds, which enhances evaporation.

### 2.1.2 Atlantic Data Set

The Atlantic data set is the output of a Lagrangian LES (Kazil et al., 2021, simulation B$_1$). The simulation covers two daytime periods and simulates the transition from an overcast closed-cell stratocumulus cloud deck to a broken, open-cell cloud deck in a pocket of open cells (POC) sampled during the Cloud-Aerosol-Radiation Interactions and Forcing (CLARIFY) campaign (Abel et al., 2020; Haywood et al., 2021). The closed-cell cloud deck at the start of the simulation is depicted in Fig. 4a while Figs. 4b, 4c and 4d illustrate the transition to an open-cell morphology. Figure 4e visualizes the 3D radiance calculations for the highlighted 6.4 km x 6.4 km sub-domain. The LES was driven by the European Centre for Medium-Range Weather Forecasts (ECMWF) reanalysis meteorology (ERA5). Its 76.8 km x 76.8 km domain follows the trajectory of the boundary layer flow,

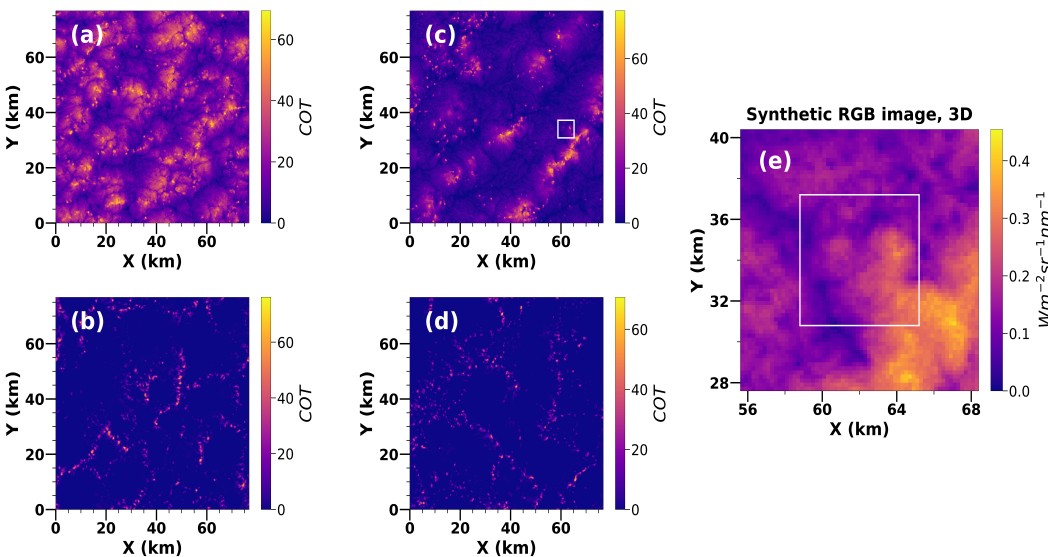

**Figure 4. a-d**: Lagrangian LES COT fields (600 nm) from the Atlantic taken during the CLARIFY campaign simulating closed-cell stratocumulus cloud deck transitioning to a broken open-cell cloud deck over a 78 km x 78 km domain; **e**: 3D Synthetic radiance calculations (Red = 600 nm), shown for the 6.4 km x 6.4 km highlighted white box sub-domain in (c).

determined with the Hybrid Single Particle Lagrangian Integrated Trajectory Model (HYSPLIT, Stein et al., 2015). On the first day of the simulation, the stratocumulus cloud deck is in the overcast, closed-cell state, with a cloud fraction near unity and negligible surface precipitation. An increase in rain water path towards the evening leads to sustained precipitation over the course of the night, accompanied by the transition to the open-cell stratocumulus state, which persists during the second day. The simulation was evaluated with satellite (Spinning Enhanced Visible and Infrared Imager, SEVIRI) and CLARIFY aircraft in-situ data. It reproduces the evolution of observed stratocumulus cloud morphology, COT, and REF over the two-day period of the cloud state transition from closed to open cells, and captures its timing as seen in the satellite imagery. Cloud microphysics was represented with a two-moment bin scheme, which reproduces the in-situ cloud microphysical properties reasonably well. A biomass burning layer that was present in the free-troposphere resulted in negligible entrainment of biomass burning aerosol into the boundary layer, in agreement with the CLARIFY in-situ measurements. Further details on the simulation, its setup, and the results are given by Kazil et al. (2021).

## 2.2 Radiance Calculations

Both sets of LES calculations contain the 3D distributions of cloud water mixing ratio ($q_l$), REF, water vapor mixing ratio ($q_v$), temperature ($T$), pressure ($p$), and other meteorological variables. From those 3D fields, the LWC is calculated as

$$LWC = q_l * \rho_{air} \tag{5}$$

where $\rho_{air}$ is the density of the ambient air, and the extinction coefficient is obtained as

$$\beta_{ext} = \frac{3 \cdot Q_{ext} \cdot LWC}{4 \cdot \rho_l \cdot REF} \tag{6}$$

where $\rho_l$ is the density of liquid water and $Q_{ext}$ is the extinction efficiency, approximated as 2 in the geometric optics regime. Furthermore, the single scattering albedo is set to 1 because all calculations are done in the visible where cloud drop absorption is negligible. In this exploratory study, the scattering by cloud drops is represented by a simple Henyey-Greenstein (HG) phase function with an asymmetry parameter of 0.85. For our purposes, it is convenient to use this fixed phase function as a proxy for the real phase function, in part because our CNN does not retrieve REF. However, the true REF distribution along with the associated Mie phase function variability are expected to introduce additional radiance variance, which will need to be considered in future, real-world CNN applications.

The radiance calculations were performed for the native-resolution 3-dimensional $\beta_{ext}$, as well as for the horizontally coarsened fields (Sect. 2.1), using the Education and Research 3D Radiative Transfer Toolbox (EaR[3]T, Chen et al., 2022) for a wavelength of 600 nm. EaR[3]T provides high-level interfaces in the Python programming language that automates the process of running 3D RT for measured or modeled cloud/aerosol fields. It builds on using publicly available 3D radiative transfer models (RTMs) including MCARaTS (Iwabuchi, 2006), SHDOM (Evans, 1998), and MYSTIC (Mayer, 2009) as 3D radiative solvers. In this study, we used MCARaTS as the solver. The calculated radiances serve as synthetic radiance observations of a hypothetical satellite imager at an orbital altitude of 705 km with a viewing zenith angle of 0° (nadir) for a SZA of 30° and solar azimuth angle (SAA) of 0°. The corresponding cloud fields were represented by the vertically integrated $\beta_{ext}$, i.e., the column COT from the LES as ground truth for the CNN.

Additional input parameters for the 3D-RT calculations include the incident solar spectral irradiance (Coddington et al., 2008), and a spectrally flat surface albedo of 0.03 with a Lambertian reflectance. Similar to the simplified representation of the cloud drop scattering, the surface reflectance assumptions are only meant to be a proxy for more complex conditions in the real world. The optical properties of 1D atmospheric components were obtained based on the U.S. standard atmospheric profile from Anderson et al. (1986) that contains a vertical distribution of atmospheric gases (e.g., $CO_2, O_2, H_2O$, etc.). We used the correlated-k absorption approach introduced by Coddington et al. (2008), which was optimized for a moderate spectral resolution radiometer. The molecular scattering optical thickness of the atmosphere is calculated based on the algorithm developed by Bodhaine et al. (1999). For each simulation, three runs were performed with 2 x $10^9$ photons each, which allows one to estimate photon (statistical) noise along with the mean radiance fields. Following the calculations, 64 x 64 pixel COT and radiance training pairs are sub-sampled from the larger generator field.

## 2.3 Pre-Processing

Before the LES-generated COT images are used as ground truth, a series of pre-processing steps are performed. A clear distinction between our approach and the one proposed by Masuda et al. (2019) is that we treat COT retrieval as a segmentation problem instead of regression. By using segmentation, we reduce the problem to a classification task where the objective is to classify a pixel into a class. Such an approach aims for accuracy over a finite $N$ set of values instead of a continuous distribution. The $N$ discrete classes represent bins over the range of COT values. By binning the true COT image, a discrete mask is obtained. To create this COT mask, we apply a lookup table to the COT image. If a pixel lies in a COT bin interval, the pixel is assigned a numerical value (or a class) corresponding to that COT bin. For instance, if the COT of a pixel is less than 0.1, it is assigned to class 0. If the COT lies in the interval $[0.1, 0.2)$, it is assigned to class 1 and so on for a total of $N$ classes. Since the bins are non-overlapping intervals, a pixel can only belong to one COT class. We note that by binning the COT, some precision is lost, but the reduction in complexity of our model via the U-Net architecture (and therefore faster training) makes up for it.

After binning, each 480 x 480 scene is divided into 64 x 64 patches or sub-domains using a stride of 32 pixels in both the horizontal and vertical directions to increase the number of samples. We also crop out the edge pixels from a scene before dividing into sub-domains. Therefore, a 480 x 480 scene can generate between 600 and 1,200 samples each of size 64 x 64 pixels depending on how many edge pixels are cropped out. Before this data is fed to train the network, we "one-hot encode" the images. One-hot encoding can be viewed as a mapping technique where a pixel is mapped from an integer value to a binary (0 or 1) class. The mapping is 1 if the pixel belongs to the class, and 0 otherwise. The COT mask becomes 3-dimensional where the depth channel is the class i.e., it goes from dimensions of 64 x 64 to 64 x 64 x $N$. If a pixel belongs to class 3, the depth-wise array of size $N$ would be (0, 0, 0, 1, 0, 0, 0, 0, ... 0) with $N-1$ zeros. In our case, we set $N$ to 36. Figure 5 illustrates this, where Fig. 5a shows a discretized mask of the true COT (obtained from LES) with a 48 km x 48 km resolution. A white box highlights a 6.4 km x 6.4 km sub-domain that is extracted. This sub-domain, shown in Fig. 5b, is then translated to a 3-D image with binary masks stacked along the channel depth axis using one-hot encoding. The resulting encoded image is shown in Fig. 5c. No pre-processing is performed on the input radiance images. Hereafter, we refer to the 480 x 480 (48 km x 48 km resolution) images generated by the LES as "scenes".

## 3 Architecture & Methodology

### 3.1 Machine Learning Terminology

The CNN is responsible for learning features and patterns that can fit the non-linear relationship between the radiance and the COT. When a radiance image is fed to the model, it gets passed through various layers, undergoing non-linear transformations and changes to size and dimension, until after the final layer when it is compared with the ground truth COT to compute the cost or error. This section will detail the inner workings of the CNN including its setup. But first, we will discuss some of the

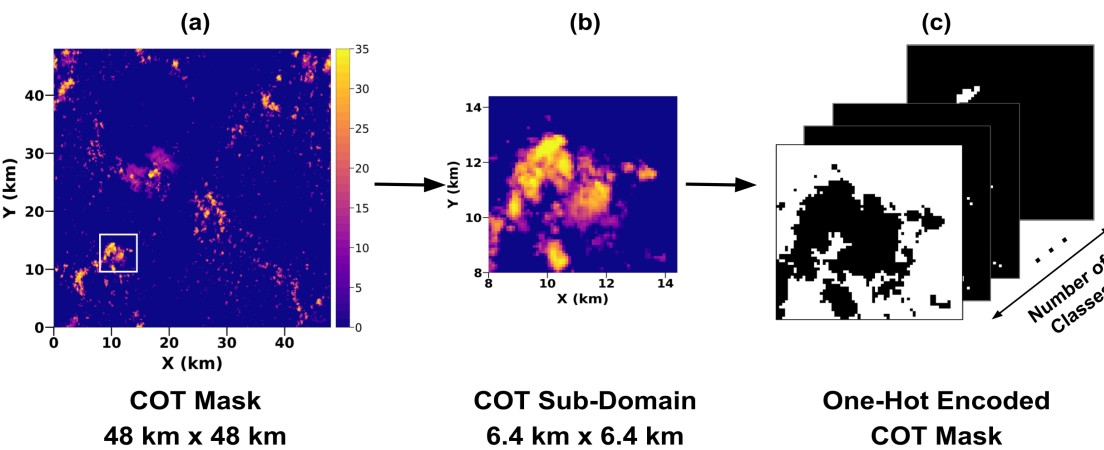

**Figure 5.** One-hot encoding translates a 2D discrete COT mask to a 3D image with $N$ binary masks. **(a)** A 48 km x 48 km discrete COT mask. The white box highlighted is a 6.4 km x 6.4 km sub-domain, which is shown in **(b)**. **(c)** shows the one-hot encoded COT that has the dimensions (64 x 64 x $N$) where a pixel viewed depth-wise (along the channel axis) would have $N$ classes of COT. Each layer represents a bin interval of the range of COT. For instance, the white pixels in the top layer have COT $< 0.1$. Similarly, in the second layer, the white pixels are those whose COT lies in the interval $[0.1, 0.2]$. The final layer's white pixels have COT $> 100$. In our case, $N = 36$

nomenclatures that will be used throughout the rest of the paper.

CNNs, at their core, are feature extractors. In our case, the goal is to learn the underlying low-dimensional and high-dimensional spatial features in the radiance imagery that when optimized, make up a mapping function to retrieve the COT. In order to extract these features, CNNs employ spatial convolution. Each convolution operation works by moving a sliding window or "kernel" over the input to produce a convolved output or "feature map". Every time the kernel is varied, the features it extracts also vary. A kernel is simply a 2D matrix that stores the coefficients or "weights" to be convolved with the input. To

put this mathematically, let the weight coefficients in the 2D kernel be represented by $w$. If the kernel size is $K$x$K$ (meaning $w$ is a $K$x$K$ matrix), and is convolving over an input image $x$ of size $M$x$M$, then the convolution operation, in its simplest form, can be written as:

$$z_{u,v} = w * x_{u,v} + B = \sum_{i=0}^{K-1} \sum_{j=0}^{K-1} w_{i,j}\, x_{u-i,v-j} + B \qquad (7)$$

where $B$ is a bias value. Biases are constants that are used to offset the output of the convolution to help reduce the variance

and provide flexibility to the network. Convolution computes the dot product over each pixel of the input over a $K$x$K$ window, offset by a bias value to obtain a single value of the 2D feature map matrix, represented here as $z_{u,v}$. The kernel then slides over the input in a horizontal or vertical direction, and repeats the operation in Eq. (7) until all the input pixels have undergone convolution. In doing so, the convolution builds and fills out the 2D feature map. The weight coefficients $w$ can therefore be

viewed as operators on input $x$ extracting a feature map $z$. Once the features $z$ are extracted, they now need to be used and combined to arrive at the mapping to the desired output (in our case, COT). Since the mapping from radiance to COT space is non-linear, an activation function $f$ (commonly used in other ML models as well) is applied element-wise to the 2D feature matrix $z$. The activation function helps the CNN learn complex patterns by only activating certain features that are most helpful in approximating the mapping to COT. The resulting output is termed an "activation map". Following Eq. (7), this activation map is given as:

$$y = f(z) = f(w * x + B) \tag{8}$$

We note that throughout the rest of this paper, we use the terms "activation map" and "feature map" interchangeably. Common activation functions include the sigmoid and tanh functions. For our proposed CNN, we use a type of activation function called ReLU or Rectified Linear Unit that only activates those features that are positive. As mentioned earlier, $f$ is applied element-wise to the 2D feature matrix $z$ and can be written as:

$$f(z) = max(0, z) \tag{9}$$

The features that a kernel extracts could be as simple as a horizontal or vertical gradient, or more complex and high-dimensional. When a number of kernels are stacked channel-wise, they are called "filters". The major advantage of using a CNN is that we do not need to know the weight coefficients of these kernels or filters beforehand; it learns these values through optimization over time. This period during which the CNN learns how to best set the weights that will result in the lowest error in its estimation of COT is termed "training". Background on the learning process is detailed in Appendix A. Additionally, we normalize the output of convolution to a fixed mean and variance using a normalization layer as is customary in machine learning. This is done before or after the activation function is applied, and helps the model to converge faster because it constrains the value range of the features, thereby stabilizing the learning process.

## 3.2 Architecture

Our CNN can be explained in two aspects: (1) the architecture and (2) the training. The architecture is derived from an existing U-Net design (Ronneberger et al., 2015). Figure 6 shows a schematic of the architecture. We opted for a U-Net style architecture for three main reasons: 1) the model complexity is lower than other architectures which thereby increases computational speed during both training and evaluation (inference); 2) the use of concatenation layers that link features learned by different stages helps the model learn new features without increasing layer depth; and 3) the U-Net has been proven to be a state-of-the-art model for segmentation problems, especially in the medical field (Litjens et al., 2017). The U-Net architecture in Fig. 6 can be broadly thought of as having two distinct halves in the U-shape: a contracting branch on the left that can be viewed as an "encoder", and an expanding branch on the right that can be viewed as a "decoder". The encoder progressively reduces or "contracts" the spatial dimensions while increasing the feature dimension. This is because the objective of the encoder is

to translate the features of the radiance imagery into a low-resolution, high-dimensional representation of the radiance (at the bottom of the U-shape). This representation is the result of learning the patterns of cloudy and non-cloudy regions at multiple scales. The decoder, on the other hand, does the opposite by building the spatial dimensions back up to its target size while reducing along the feature dimension. It projects the low resolution features learned by the encoder back to the pixel space to classify and segment each pixel into a COT bin.

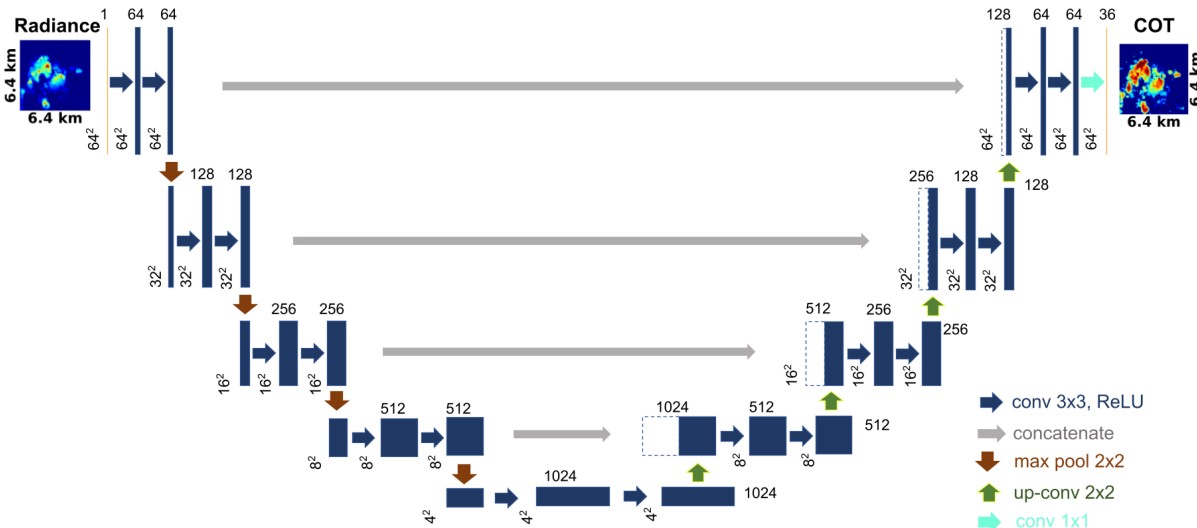

**Figure 6.** Architecture of the proposed model, based on U-Net (Ronneberger et al., 2015). The radiance images of size 64 x 64 are fed through an input layer (shown in yellow on the left). The blue rectangular blocks depict convolutional layers. The number of filters for each convolutional layer is listed above each block (64, 128, 256,...). The blue arrows depict two sequential operations - the ReLU activation and Batch Normalization. Red arrows are max-pooling layers that downsample the previous layer by half. The gray arrows represent the concatenation operation where a source feature map from the contracting path (encoder) on the left is concatenated to a feature map from the expanding path (decoder) on the right (shown as dashed blue lines). The green arrows represent upsampling via bilinear interpolation and transposed convolution. The turquoise arrow is the final 2D convolutional layer that translates the previous layer to the requisite number of output COT bins, in this case, 36. The yellow rectangle on the right is the final output layer that is activated by the softmax function to obtain 36 probabilities for each of the 36 COT bins.

### 3.2.1 Contracting Path

The contracting path (the encoder) is composed of a series of convolutional blocks separated by pooling layers. Each convolutional block has two sequential identical sets of a 2D convolution layer, normalization layer and an activation layer in that order (Convolution → Normalization → Activation → Convolution → Normalization → Activation). With each passing convolution layer, the aim is to learn a set of distinctive features of the radiance imagery that can help the model approximate

a mapping function from the radiance to the COT. Since each convolution covers multiple neighboring pixels at a time, the CNN can gather an understanding of the different distributions of radiance at different regions. The resulting outputs from each convolution i.e., the feature maps are fed forward as the input to the subsequent layer so that none of the features are learned in isolation. As we discussed earlier, since the goal of the encoder is to learn a low resolution, high dimensional representation of the radiance, the horizontal and vertical dimensions are decreased after each convolutional block through downsampling

while the number of features are increased by using more convolutional filters. The downsampling operation in the encoder is accomplished through a layer called max-pooling layer. Using the output of the convolutional block, max-pooling drops half the pixels and retains only the sharp pixels. In terms of dimensions, the model takes in an input whose dimensions are 64 x 64 x 1 (a single 600 nm radiance channel image of size 64 x 64 spanning 6.4 km x 6.4 km) while the output of the encoder at the bottom of the U-shape produces a feature map of size 4 x 4 x 1024. More details about each individual layer in the encoder are

provided in Appendix B.

### 3.2.2   Expanding Path

The expanding path (the decoder) is the right half of the architecture. It is composed of a series of decoding stages and the same convolutional blocks from the contracting path. Each decoder stage enlarges or upsamples the spatial dimensions by a factor of 2 using bilinear interpolation. For instance, after the end of the contracting path at the bottom of the U-shape, the

dimensions of the feature map are 4 x 4 x 1024. After we upsample, the new dimensions become 8 x 8 x 1024. However, interpolation is not intrinsically learnable because it does not rely on an adaptable kernel that updates during training. To add a learnable element to upsampling, an operation called "transposed convolution" is applied after interpolation. It performs the inverse of a standard spatial convolution operation to broadcast the interpolated output to a feature map containing half the number of feature maps compared to the interpolation step. Transposed convolution is further explained in Appendix C.

A core contribution of the U-Net architecture proposed by Ronneberger et al. (2015) was the use of concatenation (depicted as grey arrows in Fig. 6). This concatenation operation between layers (often referred to as "skip connections" in machine learning literature) helps to add extra information to the upsampling stage from the encoder side. It also works to reinforce some of the features learned from the original radiance imagery that may have been lost during downsampling. We then pass the concatenated feature maps through a spatial convolution block whose configuration is the same as the ones used in the

encoder. We repeat this interpolation, transposed convolution, concatenation and spatial convolution process until we reach the desired spatial size of 64 x 64. Once the original resolution is reached, a final convolutional layer with 36 filters (36 corresponds to the number of classes or COT bins) is applied, resulting in an output of 64 x 64 x 36. At this point, the output would give us the raw mapping function to retrieve the COT. However, as we discussed in Sect. 2.3, the ground truth COT is actually not the raw COT generated from the LES. Instead, it is a one-hot encoded COT of dimensions 64 x 64 x 36 where each of the 36 values

is binary (0 or 1). Therefore, we need to apply a transformation to the output of the final layer to translate it to a multinomial probability distribution across the same 36 COT bins as the ground truth. To obtain this distribution, we use the so-called softmax function (see Appendix C). It works by taking the 36 raw values of each output pixel as input and normalizing it into a PDF containing 36 probabilities proportional to the exponential of the raw values, shown in Eq. (C1). Because we use discrete

bins of COT rather than continuous COT, our approach solves a segmentation problem, rather than a regression problem, in the nomenclature of computer science. This distinction means that our CNN is simpler, smaller, and trains faster than previous architectures (e.g., Masuda et al., 2019, a regression approach). Details about each layer, activation functions (including the softmax function), and hyperparameters are explained in Appendix C.

## 3.3 Loss Function: Focal Loss

The loss function serves an important role in optimizing a machine learning algorithm. It is the method by which the model learns to minimize the difference between the ground truth and its prediction made through learned parameters. Mean squared error and mean absolute error are commonly used as loss functions in regression-based approaches. In our case, due to the use of segmentation/classification, we rely on a cross-entropy-based loss called *focal loss* (Lin et al., 2017). But, before discussing focal loss, we introduce cross-entropy loss. Also called log loss, cross-entropy measures the difference between the estimated probability and the true probability. Cross-entropy (CE) is given by:

$$CE = -\sum_{i \in I} \sum_{c \in C} p_{i,c} \cdot log(\hat{p}_{i,c}) \tag{10}$$

where $I$ is the set of all pixels, $C$ is the set of all classes, $p_{i,c}$ is the true probability of pixel $i$ belonging to class $c$, $\hat{p}_{i,c}$ is the estimated probability of pixel $i$ belonging to class $c$. We recall from Sect. 2.3 that since we use one-hot encoding for the ground truth COT, and we know that a pixel can only belong to one class, the true probability is binary. In other words, a pixel either belongs to a class (COT bin) or it does not. This can be written mathematically as:

$$p_{i,c} = \begin{cases} 1, & \text{when pixel } i \text{ belongs to class } c \\ 0, & \text{otherwise} \end{cases} \tag{11}$$

Since the softmax activation in the final layer generates a normalized PDF, the sum taken over the estimated probabilities of each class is always 1.

$$\sum_{c \in C} \hat{p}_{i,c} = 1 \tag{12}$$

Cross-entropy works well with binary as well as multi-class targets. However, cross-entropy fails to deal with class imbalance. This is because all contributions from all classes are summed together equally using only the truth and estimated probabilities without a weighting factor that can change the importance of a particular class. In our case, the imbalance between the cloudy pixels and non-cloudy pixels is particularly high with the latter sometimes occupying more than 80 % of the image. If we were to use cross-entropy as our loss function, any large errors in the estimation of the cloudy pixels would get overwhelmed or averaged out by high volumes of low errors for the non-cloudy pixels, therefore driving the cost misleadingly

low. That would lead the model to interpret wrongly that it is learning all the classes equally well when in fact it might not be. To counteract this, we use focal loss, a variant of cross-entropy designed specifically to be used in problems involving a class imbalance in the data set. Focal loss (FL), adapted from Lin et al. (2017), is given by:

$$FL = -\sum_{i \in I} \sum_{c \in C} \alpha_t (1 - \hat{p}_{i,c})^\gamma \cdot p_{i,c} \cdot log(\hat{p}_{i,c}) \tag{13}$$

Focal loss uses a "modulating factor" $(1 - \hat{p}_{i,c})^\gamma$ to address the issue of imbalance in cross-entropy loss by up-weighting misclassified examples (classes that do not have high probabilities), and down-weighting well-classified ones (classes that have high probabilities). $\gamma$ is called the "focal parameter" and acts as a smoothing factor by exponentially scaling the importance of a class. It is usually set to 1 or 2 (if set to 0, focal loss becomes cross-entropy loss). To demonstrate the effect of focal loss, let us consider an example. Let us say the model is learning a particular class "well" i.e., the estimated probability is high, say $\hat{p}_{i,c} = 0.9$ and the corresponding ground truth probability $p_{i,c}$ is 1. And let us set the focal parameter $\gamma$ to 2. Then the modulating factor would produce $(1 - \hat{p}_{i,c})^\gamma = (1 - 0.9)^2 = 0.01$. This means that the loss contribution by that class is scaled down by a factor of 100 when using focal loss instead of cross-entropy loss. By down-weighting the clear-sky pixel contribution, we can focus on improving the retrieval of cloudy pixels. In addition, the $\alpha_t$ term serves as an adaptable weighting factor. We make the weighting factor dependent on the true binary COT pixel probability $p_{i,c}$:

$$\alpha_t = \alpha p_{i,c} + (1 - \alpha)(1 - p_{i,c}) = \begin{cases} \alpha, & \text{when pixel } i \text{ belongs to class } c \\ 1 - \alpha, & \text{otherwise} \end{cases} \tag{14}$$

and we set $\alpha$ to 0.25 as recommended by Lin et al. (2017). It is worth noting that focal loss up-weights any class yielding low probabilities rather than the frequency of occurrence of that class making it more robust to any class imbalance. This ensures that the model relies on what it has learned so far, using the modulating factor to scale and correct itself.

### 3.4 Retrieval Performance Quantification

To quantify the retrieval errors, we use the pixel-centric relative root-mean-squared error (RMSE) or $R$:

$$R = \sqrt{\frac{1}{n_x \cdot n_y} \sum_{i=1}^{n_x} \sum_{j=1}^{n_y} \left( \frac{\tau_{ret}(i,j) - \tau_{true}(i,j)}{\tau_{true}(i,j)} \right)^2} \times 100\% \tag{15}$$

where $\tau_{true}$ and $\tau_{ret}$ denote ground truth and the retrieval, and $n_x$ and $n_y$ define the size of the analyzed sub-domain. In most of our examples, since we use a 64 x 64 COT image spanning 6.4 km x 6.4 km, $n_x = 64$ and $n_y = 64$. The mean (square) deviation of the pixel-level retrieval from the truth is quantified relative to the pixel-level ground truth in percentage. RMSE is a quantification of the scatter that we discussed earlier in the context of Fig. 2. For the case in Fig. 2, the relative RMSE ($R$) of

the IPA retrieval is 60.8 %.

Instead of pixel-centric metrics, one can also focus on the domain-wide retrieval performance based on the linear regression between the ground truth and the difference between the retrieval and the ground truth,

$$\tau_{ret} - \tau_{true} = a \cdot \tau_{true} + b \tag{16}$$

Typically, $a < 0$ and $b > 0$. A slope $a = 0$ and an intercept $b = 0$ would indicate a perfect retrieval in terms of the sub-domain as a whole. Unlike $R$, which also encompasses pixel-level retrieval noise, slope and intercept only capture the average deviation of $\tau_{ret}$ from $\tau_{true}$ as a function of the ground truth itself. As we noted earlier, our proposed CNN significantly reduces the bias characterized by the slope $a$ metric. However, it does not necessarily show the same extent of improvement over the IPA for the scatter (variance) characterized by $R$. This is expected as we do not directly optimize the CNN for the $R$ metric. In addition to the retrieval performance metrics introduced here, alternate metrics can be defined in terms of the two-stream transmittance as a function of COT, $log$ COT, or on the power spectrum of COT. Note that BL95 used slope and offset in $log$ COT space, and determined the slope as a function of cloud geometric thickness to introduce the first 3D COT corrections known in the literature.

For the case shown in Fig. 2, for the IPA retrieval (blue scatter), the linear regression slope (with the true COT subtracted from the retrieved COT) $a$ is - 0.79, and the intercept $b$ is 0.15. The neutral COT, $-b/a$, (0.19 in this case) is the optical thickness value above which COT is underestimated and below which it is overestimated. For our example in Fig. 2, the IPA retrieval assigns a COT of 10 to a true COT of 40, whereas a COT of 2 is retrieved as being closer to 4. Such large retrieval biases on the pixel level are much less pronounced in domain-averaged cloud properties. Equation (18) shows how the domain-average bias $\delta\tau$ can be quantified. Using the linear regression slope and intercept from Eq. (16), we can use the true COT to obtain pixel-level bias $\delta\tau(i, j)$, as shown in Eq. (17). Then, we add the pixel-level biases and take the average over the sub-domain, which yields the domain-average bias $\delta\tau$. One could also obtain $\delta\tau$ by directly using the slope, intercept and the mean of the true COT over the sub-domain, as shown in Eq. (18).

$$\delta\tau(i, j) = a \cdot \tau_{true}(i, j) + b \tag{17}$$

$$\delta\tau = \frac{1}{n_x \cdot n_y} \sum_{i=1}^{n_x} \sum_{j=1}^{n_y} \delta\tau(i, j) = a \cdot \left( \frac{1}{n_x \cdot n_y} \sum_{i=1}^{n_x} \sum_{j=1}^{n_y} \tau_{true}(i, j) \right) + b \tag{18}$$

However, the domain-average bias does not completely disappear even for larger domain sizes, which makes it a significant factor for global assessments of the shortwave surface cloud radiative effect (e.g. Kato et al., 2018), which are based on cloud transmittance calculations with imagery products as input. For the case in Fig. 2, the $\delta\tau$ is - 0.65, with the negative sign implying an underestimation.

## 3.5 Training

During training, the CNN learns from the training set, attempting to learn a non-linear mapping function between radiance and COT. At the same time as training, the CNN is also confronted with formerly unseen data that is reserved for "validation". As is common practice in machine learning, we split the data set into 80 % training and 20 % validation data. The training and validation process is repeated until the cost no longer improves at which point we declare the model to have "converged".

To train all our CNNs, we use a type of optimization algorithm called mini-batch gradient descent where we divide our training set into $K$ "mini-batches", each containing a fixed subset of the training examples. The network only sees one mini-batch of images at a time and calculates the error and mean gradient over that mini-batch. The error is backpropagated and the parameters (weights and biases) are updated before the next mini-batch is fed. Once all the $K$ mini-batches of images have been seen by the network in both forward and backward propagation directions, the network is said to have completed 1 epoch. Over a number of epochs, the CNN learns to optimize for the loss function and the error no longer decreases over time at which point we declare the model to have converged. More information about the learning and training process is detailed in Appendix A. To prevent overfitting, we stop training early when there is no significant improvement in the validation loss after a certain time. We pay special attention to the validation loss by using it as the monitored metric because it is a good indication of model generalization, our overall goal. Additionally, we use decaying learning rate to reduce the learning rate whenever learning stagnates or plateaus. We save the model weights only when there is an improvement from the previous best validation loss. L1 regularization is applied to all the convolutional layers to stabilize and improve learning by penalizing drastic weight changes.

## 3.6 Post-Processing

After the model is trained, the model needs to *predict* the COT using images of radiance. However, using the model weights that were saved during training, when a radiance image of size 64 x 64 pixels is fed, the CNN does not actually output a COT image of the same size. Rather, it estimates a probability distribution function (PDF), where each pixel $i$ has 36 probability values $\hat{p}_{i,c}$ corresponding to each class or COT bin $c$. We need to translate the PDF to an image in COT-space where we can evaluate the actual performance of the model. To accomplish this step, we use a *weighted sum* approach. For a pixel, we use the 36-value PDF estimated by the CNN and compute a product of each probability value with a COT bin interval average value $\bar{d}_c$. The $\bar{d}_c$ value is an element of an array $\bar{d}$ that spans 36 classes that is obtained using the average values of each COT bin interval used during the pre-processing step. For example, during the binning process, any COT values between 35 and 40 would be binned as class 27. The estimated probability value for each pixel at class 27 ($\hat{p}_{i,27}$) would be weighted with $\bar{d}_{27} = 37.5$, which is the average COT value of 35 and 40. This product is then summed and repeated for each pixel PDF, resulting in a 2-dimensional COT image of size 64 x 64. The predicted or retrieved COT for a pixel, denoted by $\tau_{ret_i}$ can be written as

$$\tau_{ret_i} = \sum_{c \in C} \bar{d}_c \cdot \hat{p}_{i,c} \tag{19}$$

## 4  Evaluation & Results

With the setup explained above, we evaluate how the model behaves when trained on different permutations of cloud morphology and aspect ratio. This allows us to observe how the CNN reacts to different situations, thereby providing an indication of its strengths and weaknesses. We use the metrics detailed in Sect. 3.4 to assess the performance of the CNN and compare it with the IPA retrieval. We express the performances of the CNN models and IPA as a function of three cloud metrics - cloud fraction (CF) percentage, mean cloud optical thickness (COT), and cloud variability (CV). To calculate these cloud metrics on the abscissa for each figure in this section, we only consider pixels whose COT is at least 0.1. This is done because in our binning method, we treat pixels whose COT is less than 0.1 as clear-sky and the rest as cloudy pixels. Cloud fraction percentage is the percentage of cloudy pixels in the image. The cloud optical thickness shown in the figures of this section depicts the mean COT taken across the cloudy pixels. Cloud variability is the ratio of the standard deviation of cloudy pixels to the cloud fraction. In the figures that compare the scatter performances of CNN and IPA retrievals in this section, each dot represents a binned measurement obtained by taking the mean of the performance metric (relative RMSE percentage or slope appropriately) over a number of samples to infer the overall trend in performance. Bin sizes are smaller/finer in the lower ends of each cloud metric as there are a greater number of samples in these regions. We also use histograms for a particular cloud metric performance which takes into account all samples and not just the binned ones.

### 4.1  Variability of aspect ratio

| | Model A | Model B | Model C |
|---|---|---|---|
| Number of Scenes | 6 | 24 | 96 |
| Coarsening Factor | N/A | 2 x 2 | 4 x 4 |

**Table 1.** Data used to train each of the models for Sect. 4.1. Each model is trained on a distinct aspect ratio to evaluate its influence individually.

In this subsection, the goal is to evaluate the impact and influence of aspect ratio on the CNN (and its different variants). The three CNN models A, B, and C are trained on the data that have been coarsened by factors of 1 x 1, 2 x 2, and 4 x 4 (Table 1). All three CNN models and the IPA method are tested on a data set consisting of samples with 1 x 1, 2 x 2 and 4 x 4 coarsening factors. None of the samples in this test set have been seen before by any of the CNN models during training. This allows us to evaluate if a particular model does better on a particular aspect ratio, thereby giving us insight into the ability of the CNN to generalize.

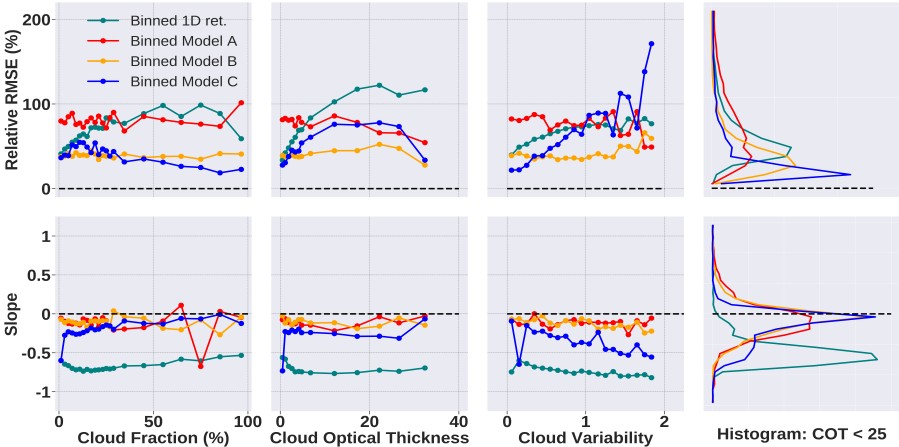

**Figure 7.** Comparison of models A, B, C (red, orange and blue lines respectively) and IPA or 1D retrieval (shown in teal) obtained by methods described in Table 1. Model A is trained on scenes that have no coarsening factors applied (1 x 1), B is trained on scenes coarsened by a factor of 2 x 2, and C is trained on scenes coarsened by a factor of 4 x 4. The dashed black line depicts the ideal retrieval. All models are being evaluated against a pool of unseen images with 1 x 1, 2 x 2 and 4 x 4 coarsening factors. The rightmost column shows the histograms samples that have COT < 25.

The first three columns from the left in Fig. 7 show the performance of the three CNN models as well as the IPA when measured against different cloud metrics (on the abscissa), and relative RMSE percentage $R$ and slope (both on the ordinate).

The histograms on the rightmost column show how much the IPA underestimates (in terms of the slope) and contains errors (in terms of $R$) for samples that have COT < 25. Among the CNN models, model A, trained on just the 1 x 1 coarsening factor data (original aspect ratio), has the highest error $R$ against all values of cloud fraction but does significantly better against variability and optical thickness. Model C, trained on the 4 x 4 coarsening factor data (quarter of the original aspect ratio), works better than model B, trained on the 2 x 2 coarsening factor data (half the original aspect ratio), when evaluated as a function of cloud fraction. However, model B performs better when measuring against mean COT and variability. Model C has differing performances when being evaluated against mean COT and variability. In the second column, as the COT increases, $R$ decreases and the slope grows closer to the ideal 0. In the third column, as the cloud variability increases, model C gets progressively worse, in terms of both $R$ and slope. But, the histogram on the right-most column shows that for COT < 25, model C is the best performing model because the mode of the $R$ percentage is closest to 0 % and mode of the slope is close to 0 (although model B is very close as well).

All three CNN models perform better than the IPA in this case study. But, among themselves, none of the CNN models seem ideally suited for all scenarios of cloud fraction, optical thickness and variability. It could therefore be inferred that a

combination of data from different aspect ratios would provide homogeneity with respect to cloud parameters as well as a wider range of spatial scales for the CNN.

 ## 4.2 Variability of cloud morphology

|  | Model A | Model B |
| --- | --- | --- |
| Number of Scenes | 1 | 5 |
| Coarsening Factor | 4 x 4 | 4 x 4 |

**Table 2.** Data used to train each of the models for Sect. 4.2. Model A learns from a single cloud generator scene whereas model B learns from 5 generator fields.

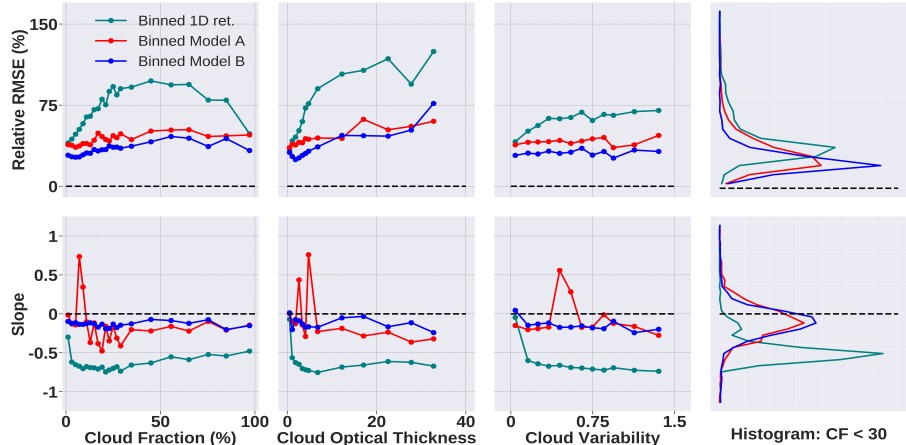

**Figure 8.** Comparison of two CNN models A and B (shown in red and blue respectively) with IPA or 1D retrieval (shown in teal) obtained by methods described in Table 2. Model A is trained on only 1 scene (coarsened by a factor of 4 x 4) whereas model B is trained on 5 scenes (also coarsened to a factor of 4 x 4). All 3 are being evaluated on samples from a holdout scene (with shear). The dashed black line depicts the ideal retrieval.

The goal of this study is to evaluate the importance of diversity in varied cloud generator fields, and if a model trained on a limited number of such fields can generalize to unseen data. We also test if a model trained on multiple fields loses accuracy over individual fields by trading for generalization. All models and the IPA method are evaluated on a holdout scene that has not been seen by any of the CNN models during training.

560

Model A is trained on samples from a single LES cloud generator field at a 4 x 4 coarsening factor. Model B is trained on samples from 5 different cloud generator fields. From Fig. 8, it is once again clear that both the CNN models, A and B, outperform the IPA retrieval consistently across all metrics. There is a clear distinction between CNN and IPA performances.

In the left most column, the IPA retrieval is the most error-prone in terms of the relative RMSE percentage across all cloud fractions, and underestimates the true COT by more than a 50 % margin in terms of slope. The same is reflected in the second column where the IPA either gets worse with increasing COT or remains off the ideal slope by a significant margin. The slope of the IPA retrieval drops off significantly with higher cloud variability in the third column while $R$ grows worse as well. With the two CNN models, model A, trained on a single scene, performs comparatively well over large portions of cloud fraction, variability and optical thickness but struggles with low COT, low CF, and certain sections of CV where it both underestimates and overestimates. This could be inferred as the inability of model A to generalize to any images that were not similar to the envelope of the original training scene because a single scene would not contain enough variability or diversity. On the other hand, model B, trained on 5 scenes, is far more stable and consistent across all 3 cloud metrics vs both slope and $R$. In other words, model B does not lose accuracy in return for better generalization. The histogram on the top right shows how the IPA is highly error-prone, with most samples having a higher $R$ percentage than either of the CNN models. This is also captured in the bottom right slope histogram with a significant shift toward lower slopes. CNN model A and model B perform comparatively well in both histograms, with the latter slightly edging out in terms of better $R$ performance. Therefore the impact of using multiple cloud generator fields is quantifiably higher and more useful for the model as it gains a more generalizable interpretation of the data.

## 4.3 Training on a sampled data set

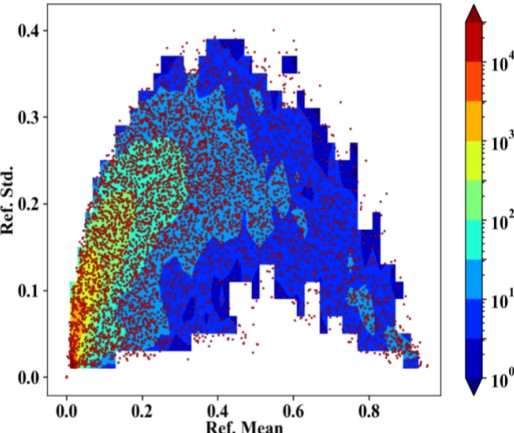

**Figure 9.** A distribution of standard deviation of reflectance ($Ref_{std}$) vs mean of reflectance ($Ref_{mean}$) for the gridded and sampled data set. Each red scatter dot represents a 64 x 64 reflectance image belonging to the consolidated gridded/sampled data set consisting of the 1 x 1, 2 x 2 and 4 x 4 coarsening factor data pools.

In this section, using knowledge gained from the previous two case studies, we build a new training data set based on the LES data from the Sulu Sea (Sect. 2.1.1). Now that we know the advantages of using multiple cloud generator scenes and coarsening, we develop a data set that combines both. However, training a model just by combining all 3 coarsening factors

from all 6 scenes will be inefficient. This is because of the numeric imbalance where the highest coarsening factor produces about 16 times the number of samples produced by the lowest coarsening factor. To overcome this imbalance and data bias,

we use a gridded approach to select a representative sample from a selected region and therefore limit the total number of samples but retain the importance in terms of the contribution to statistical diversity. We employ a sample selection technique that randomly selects data samples from three data pools of differing coarsening factors - 1 x 1, 2 x 2 and 4 x 4 at grid boxes defined by standard deviation of the reflectance $Ref_{std}$ and mean of the reflectance $Ref_{mean}$. We use these two metrics because $Ref_{mean}$ captures the mean brightness in the data set while $Ref_{std}$ represents the general inhomogeneity in the data.

The steps are described below:

1. Calculate $Ref_{std}$ vs $Ref_{mean}$ for the total of 24,000 samples coming from all 3 domains. Of those samples, about 1,200 come from the 1 x 1 domain, 5,000 come from the 2 x 2 domain and nearly 19,000 come from the 4 x 4 domain.

2. Divide the $Ref_{std}$ vs $Ref_{mean}$ distribution into grid boxes where each grid box corresponds to ranges of deviation and mean.

3. Randomly select data samples within each grid box from the three data pools.

One aspect to note is that because the total number of samples in each data pool differs, samples are more likely to be selected from the pool that contains a higher number of data points. As a workaround, to achieve a uniform probability in selection for the three data pools, we weighted the random selection in step 3 based on the total number of samples of the data pool (higher total number gets lower weights). The sample selection was performed for each grid box based on a given number

of sample selection per box defined by the user. If the given number exceeds the total number of samples within the grid box, all the data samples in the grid box will be selected. The resulting data set has 548 samples from the 1 x 1 domain, 1,522 from the 2 x 2 subset and 3,180 from 4 x 4, all of which are chosen with selected randomizations. Figure 9 shows the distribution of $Ref_{std}$ vs $Ref_{mean}$ for the resulting sampled data set. While this data set is not completely balanced despite having a more uniform $Ref_{std}$ vs $Ref_{mean}$ distribution, it is representative of the diversity in the data. The hypothesis is that a CNN trained

on this data set can retain accuracy over individual cloud fields and also generalize to unseen data, even better than the models seen in Sect. 4.1 and Sect. 4.2.

Figure 10 shows the distribution of the sampled data set across (a) cloud fraction, (b) cloud optical thickness and (c) cloud variability. We show this figure to illustrate how balancing the data set using $Ref_{std}$ vs $Ref_{mean}$ affects the other metrics.

In Fig. 10a, we see that the cloud fraction is relatively well represented by the sampled data set, although not ideal. More

than 90 % of the data is in the CF < 50 % region. Figure 10b shows a much higher level of imbalance, with half of the samples having mean COT $\leq$ 3.8. Figure 10c shows the highest degree of disparity with 90 % of the samples having CV $\leq$ 1.2. This will mean that the model will not be exposed to much diversity in variability but the fact remains that this data set is still representative of the brightness and inhomogeneity distribution.

Figure 11 shows the performance of the CNN and 1D retrievals. The CNN, trained on the gridded/sampled data set consisting

of images from 1 x 1, 2 x 2 and 4 x 4 coarsening factors from all 6 generator fields, outperforms the IPA retrieval across all

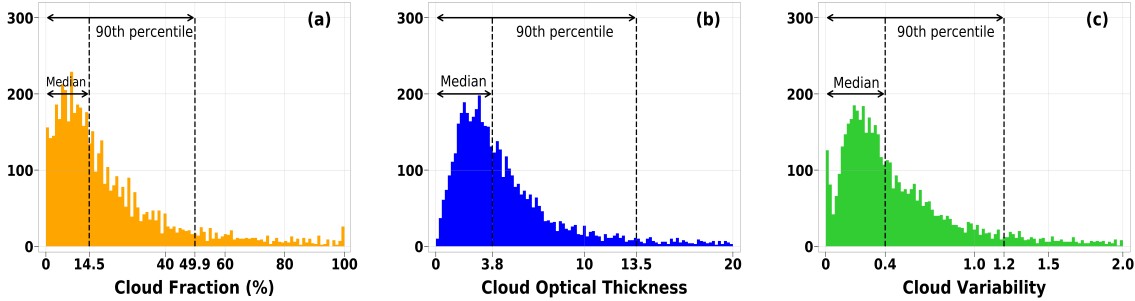

**Figure 10.** Histograms showing the distribution of the sampled data set used for training in terms of **(a)** cloud fraction, **(b)** cloud optical thickness and **(c)** cloud variability respectively. Despite being sampled using a gridding approach of $Ref_{std}$ vs $Ref_{mean}$, the balance necessarily reflected in these histograms. Particularly with the COT distribution seen in **(b)**, we see that the median COT is only 3.8 and 90 % of the samples have COT $\leq$ 13.5. The same imbalance is seen in **(c)** where the median cloud variability is only 0.4 and 1.2 is the 90[th] percentile point.

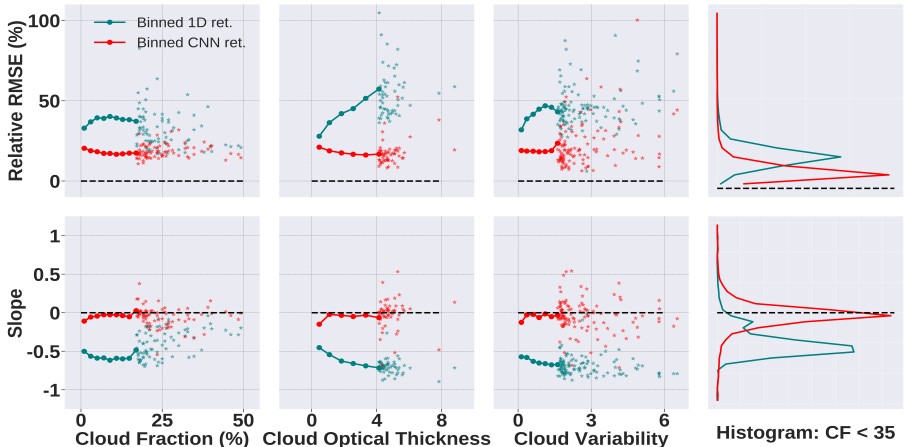

**Figure 11.** Comparison of CNN (shown in red) and IPA or 1D retrieval (shown in teal) obtained by methods described in Sect. 4.3. The CNN is trained on the gridded/sampled data set consisting of images from 1 x 1, 2 x 2 and 4 x 4 coarsening factor domains, and is being evaluated on a mix of unseen samples from all three domains. The dashed black line depicts the ideal retrieval.

cloud metrics against slope and error $R$. While the IPA underestimates low CF images with slopes close to -0.5, the CNN was significantly closer to the ideal retrieval. We use scatter points to depict certain samples in the higher ranges of the cloud parameters as there are too few such samples, which would sway the solid line plot unfairly. For the histograms on the rightmost column, consisting of samples that have cloud fraction $<$ 35 %, the CNN is much less error prone and performs well over the entire 64 x 64 sub-domains. Therefore, a uniform and representative mix of images (in terms of parameter space) from different domains yields better performance. Ultimately, this shows that the reduction in data set size does not negatively affect


the performance and quite on the contrary, can improve it, as long as it is done strategically. This experiment reinforces one of the objectives of our work, which is to demonstrate different training methods to identify optimal approaches.

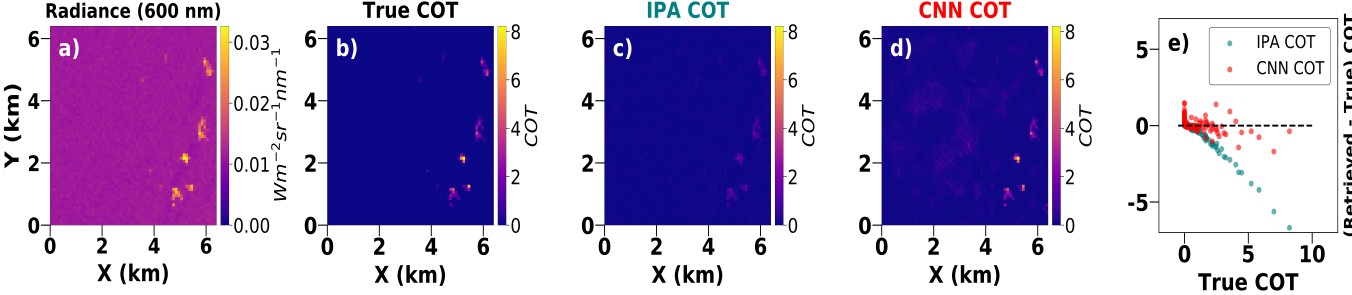

**Figure 12.** **(a)** An image of the 64 x 64 (6.4 km x 6.4 km) radiance channel (600 nm) taken from the Sulu Sea (Sect. 2.1.1) but not shown to the CNN during training. **(b)** The corresponding 64 x 64 COT. **(c)** COT as retrieved by the IPA method of the image in (a). **(d)** COT as retrieved by the CNN trained using methods explained in Sect. 4.3. **(e)** A scatter plot that compares the IPA and CNN retrievals, with the former underestimating for large COTs.

Figure 12 shows a panel of images with a scatter plot to compare the retrievals by IPA and the CNN for an unseen radiance image from the Sulu Sea. The scatter plot in in Fig. 12e shows how the IPA underestimates for medium to high COTs while the CNN remains relatively close to the ideal retrieval.

## 4.4 Testing the model on a new geographic region

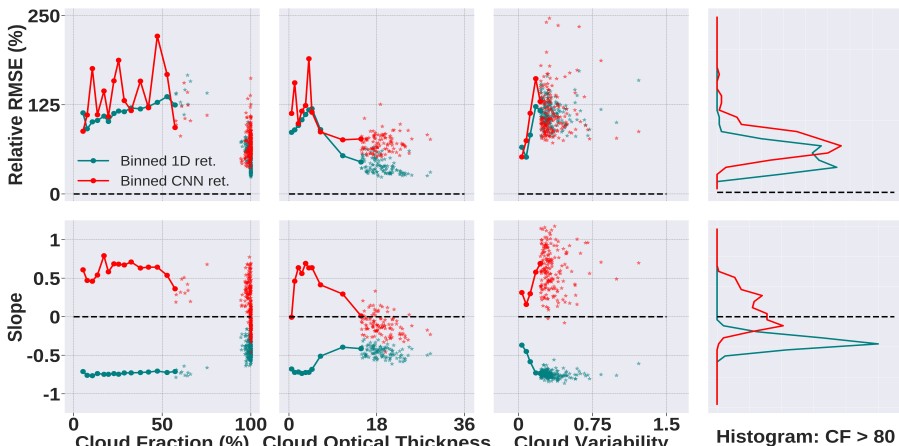

**Figure 13.** Comparison of CNN (shown in red) and IPA or 1D retrieval (shown in teal) when being evaluated on a new geographic region (Atlantic, Sect. 2.1.2.). The CNN is trained on the gridded/sampled data set consisting of images from 1 x 1, 2 x 2 and 4 x 4 coarsening factor domains. The dashed black line depicts the ideal retrieval.

Here, we present the results of the model trained on the gridded and sampled data set from the previous study (Sect. 4.3) when applied to a completely new geographic location, in this case, the Southeast Atlantic. The purpose of this application

is to observe whether the model is capable of generalizing to highly dissimilar data from a region with vastly different cloud morphologies. This is an important step to ensure that the CNN is not restricted to its training envelope and has learned the right features that can be applied more broadly. We also examine and identify the strengths and weaknesses of such an application with respect to cloud parameters.

Looking at only the abscissa ranges in Fig. 13, we can see that this is a vastly different data set. The cloud fraction percentage in the samples is high, especially compared to the training data from the Sulu Sea. The cloud variability in the data set is on the other extreme end with most samples having very low variations. When the CNN (trained on the gridded data set) is evaluated on this completely different cloud morphology, the results vary in two major ways. First, it marginally under-performs compared to the IPA in terms of $R$. The IPA is better across the top panel - the CNN has a higher $R$ for cloud fractions $< 75$ %

and for cloud variability $< 25$. The top right histogram shows as much, even for cloud fractions over 80 % where the IPA has a very low error. This is not surprising because the IPA is expected to do well in areas with high CF.

On the other hand, the slope paints a contrasting picture. Both the IPA and CNN perform underwhelmingly over low COT and low CF where they underestimate and overestimate respectively, but the CNN edges the IPA for CF $> 80$ % as shown in

the histogram. This is significant because it tells us that the CNN, which has not seen any data similar to the high CF seen here, can perform satisfactorily although not ideally. A model trained on a small and imbalanced data set is still capable of producing good results through the right training approaches.

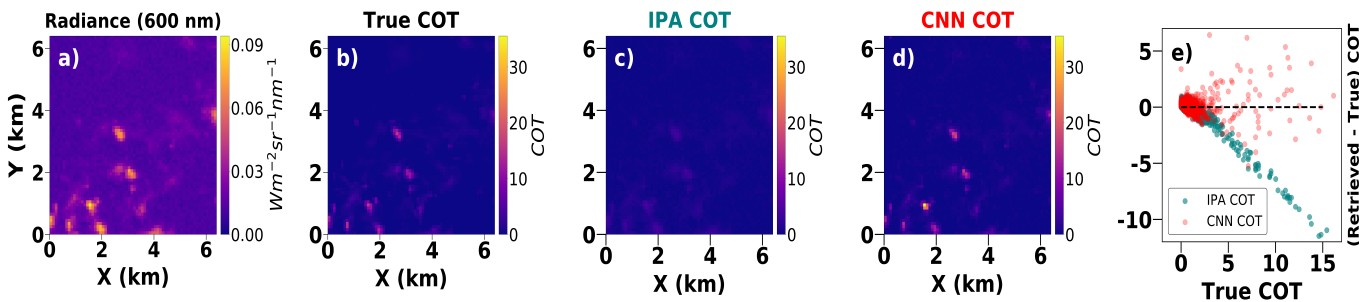

**Figure 14. (a)** An image of the 64 x 64 (6.4 km x 6.4 km) radiance channel (600 nm) taken from the Atlantic (Sect. 2.1.2) to test the CNN's ability to generalize to new geographical regions. **(b)** The corresponding 64 x 64 COT. **(c)** COT as retrieved by the IPA method of the image in (a). **(d)** COT as retrieved by the CNN trained using methods explained in Sect. 4.3. **(e)** A scatter plot that compares the IPA and CNN retrievals, with the former underestimating for large COTs.

In Fig. 14, we show a panel of images with a scatter plot to visualize the actual retrievals from the IPA method and the CNN. We use a radiance image from the Atlantic as the input and the scatter plot in Fig. 14 illustrates how the IPA, despite being

better in many regions in the Atlantic, remains erroneous by underestimating high COTs. The CNN has a high variance but performs better than the IPA for most COTs.

## 5 Summary and Discussion

In this paper, we introduced a U-Net-based, CNN architecture to infer COT fields from shortwave radiance as observed by satellite or aircraft imagers. Unlike the heritage IPA that is used almost exclusively in current operational algorithms, the CNN

takes the spatial context of a given pixel into account to reduce or mitigate retrieval biases arising from net horizontal transport (also known as 3D effects). This exploratory study, preceded by Okamura et al. (2017) and Masuda et al. (2019), is built on synthetic data at a fixed spatial resolution. Cloud fields from LES output were fed into 3D-RT calculations to simulate what an imager with a 100 m pixel size would measure, and paired with the corresponding optical thickness ground truth from the LES to train the network.


The intent of this line of research is to work towards future real-world applications of CNNs by minimizing training time while ensuring high-fidelity retrievals, even when the analyzed cloud scenes deviate from the original training envelope.

These goals were approached in two ways: (a) through the U-Net architecture itself, which has only moderate depth (number

of layers) and therefore requires less processing time for radiance/optical thickness training pairs than more complex networks; (b) by strategically limiting the amount of training data. To accomplish this, we used only six LES-generated scenes associated with varying aerosol conditions and wind shear, spanning a limited range of cloud morphologies. From these scenes, we sampled 6.4 km x 6.4 km mini-domains as training pairs (radiance and true COT) and tested the performance of the CNN on unseen data for different training data constellations.


The first experiment (Sect. 4.1) explored the impact of scale and degree of homogeneity by horizontally spreading the original LES fields by factors of 2 and 4. This spatial coarsening procedure homogenizes the cloud fields while altering the aspect ratio of individual clouds at the same time. By design, the U-Net considers net horizontal photon transport not just at the native spatial resolution of the training data, but also at a cascade of spatially aggregated versions of that data, shown in the lower

levels of the architecture (Fig. 6). CNNs that were trained on the original, 2 x 2, and 4 x 4 coarsened data all out-performed the IPA when applied to unseen data from a combination of scale levels. The retrieval fidelity was quantified via performance parameters such as the "slope" as defined above, as a function of cloud metrics such as cloud fraction and cloud optical thickness for each analyzed 6.4 km x 6.4 km mini-domain. From this experiment, we found that changing aspect ratios did not significantly alter the physics to the detriment of retrieval fidelity, despite the findings of BL95. However, we must note that there

are several points of difference between the BL95 study and ours. Most notably, BL95 used 2D Landsat imagery with mostly stratocumulus cloud fields while we use 3D LES with isolated cumulus clouds. It should also be noted that the BL95 paper varies cloud geometric thickness to change the aspect ratio while we vary the horizontal resolution keeping other dimensions

constant. Therefore, we cannot conclusively say that the aspect ratio was the sole direct cause of this discrepancy in the IPA retrieval performance.


In the next experiment (Sect. 4.2), we explored the impact of cloud morphology on retrieval fidelity by training a CNN on a single cloud morphology, and found that more diversity with respect to morphology does not negatively affect the performance of the retrieval. That is because the single-morphology CNN, applied to unseen data with that same morphology, did not perform better than its multi-morphology counterpart. On the contrary, the diversely trained CNN proved more robust, especially

in certain sub-ranges of some cloud metrics (for example, for small cloud fractions).

Therefore, the performance of single-scale and single-morphology CNNs on unseen training data of their own kind was not better than diversely-trained CNNs. Since the latter turned out to be more robust, this suggested that diverse training data should be systematically combined in an optimal CNN. To keep the training sample number low, we developed a balancing approach

to sub-select image pairs according to their location in a two-dimensional parameter space spanned by radiance mean and standard deviation, which can be regarded as proxies of mean COT and inhomogeneity, respectively. This diversely-trained, balanced CNN (Sect. 4.3) performs best compared to all the versions tested in the other experiments. The general conclusion is that strategically selected training data can lead to higher retrieval fidelity than sample-rich training data without or with improper balance with respect to parameter space. The combined parameter space as shown in Fig. 9 could be called the general

"training envelope" of the balanced CNN.

It is important to note that even the diversely-trained, balanced CNN is only diverse within the confines of the original six generator scenes. This narrow choice had been made consciously to test the limits to which training data and thus training time could be minimized. In reality, however, cloud scenes can fall well outside the training envelope – not necessarily in

terms of our simple two-dimensional parameter space of radiance mean and standard deviation, but in terms of a plethora of morphology parameters such as cloud-to-cloud distance, cloud fraction, vertical distribution, geometric tilt etc., not to mention sun-sensor geometry. One way to assess the robustness of the CNN in this regard would be to use LES data from the same set, but at a different time step and therefore a different stage of cloud evolution. We instead chose to use LES data from an entirely different region and cloud type, and tested the performance of the CNN trained with data specific to the Sulu Sea with

unseen data from the Southeast Atlantic. Overall, the CNN and the IPA performed about equally well in this case; the slightly better performance of the IPA in terms of RMSE was balanced out by the slightly better performance of the CNN in terms of slope. This alone is surprising. Since the IPA is based on physics that does not entail any learning, one would have expected it to out-perform the pattern-based CNN when encountering a previously unseen cloud scene with a completely different morphology. Even more surprisingly, the CNN out-performed the IPA in terms of slope especially for the stratocumulus sub-set

of the unseen data (cloud fraction larger than 80 %), even though this was a cloud morphology that the CNN had never seen during the training. This cloud type in particular should have been the strength of the IPA because it is less inhomogeneous than open-cell convection. For the scattered cloud scenes associated with open cell convection, the IPA underestimated COT by

about as much as the CNN overestimated it (slopes of -0.8 and 0.8, respectively), and the bias increases with cloud variability as one would expect.


The Atlantic data set represents a limiting case where the superior performance of the CNN trained with the Sulu Sea data has dropped to a level similar to or worse than the IPA reference retrieval (except for the stratocumulus sub-set). At this point, a regionally specific training based on locally initialized LES with a similar CNN architecture would become necessary. In this paper, we stopped short of re-training the CNN for another region. A related paper from Wolf et al. (2022) does train the CNN

for a different region and applies it to real-world observations from satellite imagery and flux radiometers at the surface and on aircraft. In addition, Chen et al. (2022) explores a CNN on aircraft imagery data from the Philippines region.

Aside from regional and cloud-type driven differences in cloud morphology, there are other factors that limit the immediate applicability of CNNs for operational retrievals. The most significant challenge is domain size. As described by Song et al.

(2016), net horizontal photon transport in the visible is mostly driven by COT contrasts, regardless of the physical distance over which they occur (an exception is the near-UV wavelength range where scattering by air molecules plays a role). As such, 3D effects do not stop at the domain boundary, and the CNN will lose its accuracy if the most important spatial inhomogeneities occur over scales larger than 6.4 km. It is possible to train with larger domains, but this increases the complexity and training time of the CNN. To solve this problem, one could train a CNN with a flexible, cascading hierarchy of domain sizes. The U-Net

architecture is ideally suited to generalize the approach in this fashion.

Future work needs to explore this avenue, while also accounting for the relatively coarse pixel resolutions in typical imager radiance data (often around 1 km). At these scales, many cumulus clouds are not resolved (Koren et al., 2008), but they have a collective radiative effect that must not be ignored. For such sub-resolution clouds, the CNN runs into the same limitations

as its 1D counterpart, the IPA. Here, the spectral signature of net horizontal transport between spatially inhomogeneous cloud elements could come to the rescue. This inhomogeneity-induced parameter is detectable in spectral radiances, regardless of the scale at which clouds occur, and it might become another input parameter (Schmidt et al., 2016) to a future CNN architecture that could retrieve pixel-level cloud fraction and COT.

CNNs will always be limited by the availability of realistic training data. Since it may be impractical to provide regionally specific LES-based training data everywhere on the globe, it will be necessary to use CNNs that are trained on data from one region, as proxies for others, as long as certain cloud morphology parameters are comparable. In a future paper (Chen et al., 2022), we show that radiance closure (the consistency between radiances as measured by an imager and as calculated based on a CNN or IPA retrieval) is an appropriate tool to assess retrieval performance in the absence of ground truth validation data.


To generate regionally specific training and validation data for the CNN, cloud tomography (Levis et al., 2020) might be an alternative or addition to LES, at least for some cloud types. In this approach, 3D cloud fields are reconstructed from

multi-angle radiance observations as available from some satellite radiometers without any training. This is because LES and tomography-generated training data have the additional advantage of providing the vertical distribution of the cloud extinction. Since our simple CNN only retrieves 2D COT fields without consideration of the cloud top geometry, it is important to keep track of biases associated with this simplification.

Finally, additional spectral channels, especially in the shortwave infrared, would provide access to geophysical parameters well beyond COT – for example thermodynamic phase, drop size, and parameters of aerosol residing between clouds, facilitating and improving joint quantification of cloud-aerosol radiative effects from satellite imagery, even for complex or inhomogeneous scenes. This research, along with its practical applications, is only just beginning.

## Appendix A: The Learning Process for CNNs

Learning occurs through a process called "backpropagation", short for backward propagation of error. The model estimates features during forward propagation from left to right in Fig. 6 and Fig. A1, and using a loss function, the error between the "learned" estimation of the COT and the ground truth COT is calculated. During backward propagation from right to left, this error is then propagated backwards through all the layers, and the gradient of the loss function with respect to the weight of each layer is computed using the chain rule from differential calculus. This is one of the major reasons machine learning algorithms often consume large amounts of time. CNNs often have millions of learnable parameters i.e, the weights and biases, and computing the gradient for each is a time-consuming task. In essence, the gradients inform the network of how much the weights and biases need to be varied. This is because the gradient with respect to each weight and bias is simply subtracted from the previous weight and bias value. This is called the update step. This entire process of forward propagation, error computation, backward propagation and parameter update is repeated until the model converges to a global minimum.

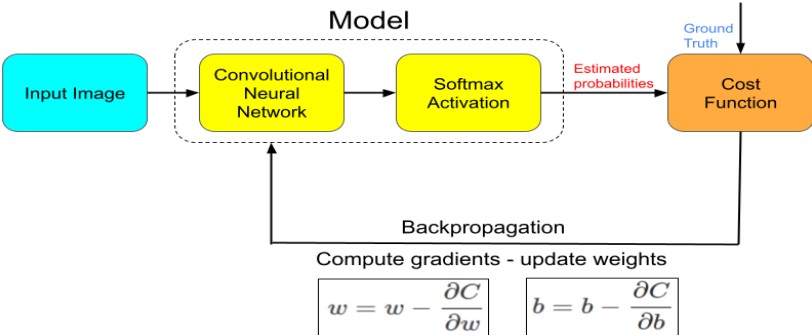

**Figure A1.** Schematic representation of the model training approach. $C$ is the cost that needs to be optimized, $w$ refers to the weights of a neuron, and $b$ is the bias. The final layer of the CNN is cast to probability space using a softmax activation function and the estimated probability at the end is compared with the true probability (that of the true COT) using a cost function which computes a cost. The cost or error is then backpropagated through the network backwards during which the weights and biases are updated. This process is repeated for different batches of images until the loss is minimized when the model is said to have converged.

We train the model using the Adam optimizer (Kingma and Ba, 2014). The Adam optimizer is an extension of the commonly used gradient descent algorithm that is used to train machine learning algorithms. In a regular stochastic gradient descent algorithm, a single constant learning rate is used across all the weights in the model throughout training. The Adam optimizer, which stands for Adaptive Moment Estimation, uses a slightly different approach to gradients. It calculates a moving average of the mean (the first moment) of the gradients, and a moving average of the squared gradients (the second moment). It uses two adjustable parameters $\beta_1$ and $\beta_2$ to control the rates at which these averages decay over time. From the Kingma and Ba (2014) paper, we can glean that this algorithm has numerous advantages, most notably that it has no memory requirements and is appropriate for noisy gradients. This optimization algorithm has come to be popular and is used as the default optimization for numerous ML problems. We initialize the optimizer with a learning rate of 0.001 but using a decaying learning rate scheduler so it can reduce the rate when the loss is not improving.

**Appendix B: Encoder (Contracting Path)**

The first convolutional block ingests an input radiance image of size 64 x 64 and produces 64 feature maps. These feature maps are the result of 64 filters convolving over the input radiance imagery to extract its features. We use a stride of 1x1 for all 2D convolution operations. Each filter produces a single 2D feature map of size 64 x 64 as we perform 2D convolution. Thus, 64 filters yield 64 2D feature maps. These resulting 64 2D feature maps are stacked channel-wise (the dimensions then become 64 x 64 x 64) and fed to the next 2D convolutional layer to extract more features. Each subsequent convolutional block doubles the number of filters (and therefore the number of features) until we reach 1,024 filters in a bid to gather enough features about the underlying data that can then translate to predicting COT for unseen radiance imagery. In the encoder, all filters use a 3 x 3 kernel and the convolution uses a stride of 1 x 1. The batch normalization layer precedes the activation and helps stabilize the training by applying a transformation to the feature maps to maintain the mean around 0 and the standard deviation around 1. All convolution layers in the encoder are activated by the ReLU activation given in Eq. (8) after normalization. Additionally, we pad the convolution operation with zeros each time to retain usable resolutions. We employ a max pooling layer in between convolutional blocks for two reasons - 1) to reduce the number of dimensions in the feature maps by downsampling along the spatial dimensions, which reduces the computation in the network; and 2) to extract the sharpest features while dropping noisy ones. We use a pooling size of 2 x 2 and use a stride of 2 x 2 to ensure that the spatial dimensions get halved.

**Appendix C: Decoder (Expanding Path)**

The decoder is in charge of using the low-resolution representations of the radiance imagery generated by the encoder to build it back to its target size through upsampling. The upsampling operation is done by bilinear interpolation in the decoder. The spatial dimensions get doubled each time this operation is performed. Immediately after interpolation, we apply a transposed convolutional layer with a 2 x 2 kernel and a 1 x 1 stride and also pad the operation. Transposed convolution (Dumoulin and Visin, 2016) works by switching the forward and backward passes of a traditional convolution and is used when going from

a lower-dimensional space to a higher-dimensional space while maintaining a connectivity pattern between the two. We use a 2 x 2 kernel with a single stride and padding to ensure the output dimensions remain the same as the input. As noted by Dumoulin and Visin (2016), it is possible to replace transposed convolution with a regular spatial convolution step but that would require additional padding thereby reducing the implementation efficiency. To map the latent space of the contracting path's output to a data distribution, it is necessary to upsample at the lowest spatial dimension to the appropriate size of the ground truth/output. The obvious way to scale up to the output dimensions is to use upsampling layers that use interpolation (e.g., bilinear, nearest-neighbors). However, interpolation is not "learnable" as it is not dependent on a kernel. Therefore, to have the model learn optimal ways of upsampling, we rely on a subsequent transposed convolution layer following upsampling. The representation generated by the transposed convolution layer is then channel-wise concatenated with the corresponding feature map having the same spatial dimension from the encoder side. A convolution block then learns new features and obtains a new representation. This convolution block is also useful to prevent so-called "checkerboard effects" that result when transposed convolution is used in isolation (Odena et al., 2016). The transposed convolution is again activated by the ReLU function after batch normalization. These operations are repeated until the spatial resolution reaches 64 x 64.

As stated earlier, since this is a segmentation approach, we need to translate the output to probabilities where each pixel has a probability distribution across the 36 COT classes. In other words, the network needs to tell us how likely a pixel is to belong to a COT bin. We accomplish this probability translation by applying a softmax function to the output. This function can be written as:

$$f(z_i) = \frac{e^{z_i}}{\sum_{j=0}^{N-1} e^{z_j}} \tag{C1}$$

where $f(z_i)$ is a function acting on a value or class $i$ in a feature map vector $z$ which contains $N$ such values (classes). In our case, $N$ is 36.

*Author contributions.* VN developed the CNN and performed the experiments and case studies, and wrote the manuscript with input from the co-authors. KSS is a PI of the CAMP[2]Ex mission, assisted with the development of the methodology, and writing and editing the manuscript. HC aided with the data generation and editing the manuscript. TY helped generate part of the data used in this manuscript and aided in its writing and edit. JK helped generate part of the data used in this manuscript and helped in its writing and edit. HI provided the software and programming code for MCARaTS, valuable expertise in machine learning, and aided in the manuscript editing. KW and GF helped in the edit of the manuscript.

*Competing interests.* The second author (Sebastian Schmidt) is a member of the editorial board of Atmospheric and Measurement Techniques (AMT).

*Acknowledgements.* The authors acknowledge support from NASA grant 80NSSC18K0146 in support of the NASA CAMP$^2$Ex mission. Vikas Nataraja was also supported by NASA grant NNX15AF62G in support of the NASA ORACLES mission. Hironobu Iwabuchi acknowledges that this study was partly supported by the 2nd Research Announcement on the Earth Observations of the Japan Aerospace Exploration Agency (JAXA; PI No. ER2GCF204).

835

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
