# Peer review of "Segmentation-Based Multi-Pixel Cloud Optical Thickness Retrieval Using a Convolutional Neural Network"

_Atmospheric Measurement Techniques, 2022_

## Author Response (AR1)

**This document contains the combined responses to RC1 and RC2. We would like to take this opportunity at the top to thank both the reviewers and the editor for helping us improve the manuscript. We hope our changes in the revised manuscript reflect this.**

**Response to RC1**

Dear Reviewer,

Thank you for your feedback on our manuscript, we appreciate you taking the time to do so. Our responses to your comments are detailed below:

Line 17-18, a few references regarding the importance of COT might be helpful, with Zhao and Garrett (2015, doi: 10.1002/2014GL062015) suggested.

-> Thank you for pointing that out. We will follow this advice in the revised version. Instead of including the haze-specific paper the reviewer suggested, we chose the more general cloud climatology paper on ISCCP by Rossow and Schiffer (1991, doi: 10.1175/1520-0477(1991)072<0002:ICDP>2.0.CO;2) where COT features prominently in Figure 4. We will change the text as follows:

"*Cloud optical thickness (COT) is important for the shortwave CRE, and it is therefore a key parameter in cloud climatologies (e.g., Rossow and Schiffer, 1991, Figure 4). Deriving the COT accurately from satellite imagery will help to improve our understanding of the energy budget.*"

Line 23, the inhomogeneity issue exists in both spatial and temporal.
-> We agree, but since we are proposing a spatial-based solution with the Convolutional Neural Network (CNN), we limited our discussions to spatial-related problems.

Line 46, IP bias is not defined yet in the main text, while defined in the abstract.
-> Thank you for pointing this out, we will make this change in time for the next stage of the review.

Line 51-53, I appreciate the information here. However, I wonder if the satellite spatial resolution is high enough to make us ensure that the optimum occurs at a scale of about 1 km.

> This is a good point. However, the optimum was determined with synthetic data (Figure 13 in Davis et al., 1997, https://doi.org/10.1175/1520-0469(1997)054<0241:TLSBIS>2.0.CO;2, cited in the paper). In this case, artificial clouds representing stratocumulus were generated with a fractal cloud algorithm, and fed into 3D RT calculations to generate synthetic radiance data. From these, COT was derived, and the optimum resolution of an imager was determined. These (and related) findings ended up guiding the choice of the spatial resolution of EOS imagers (at least they were one factor).

Line 63, "distinguished"
-> We will address this typo in time for the next stage of the review.

Figure 2, It seems to me that the difference (IPA COT) has a very good linear relationship with true COT, making me think that the IPA COT could be highly improved by simply corrected with this linear relationship. If this is true, why do not we use this simple method?
-> This is a very valid and appropriate observation. In fact, the IPA dependence of (retrieved-true) COT on the COT is even more linear than for the CNN retrieval. We will add this observation to the revised text. It is indeed possible to parameterize this effect as a 3D correction, and this has been done in the past. We described these approaches in Section 1.2. See for example Equations (2), (3), (4), and the related literature citations. Iwabuchi and Hayasaka (2002) introduced a more complex statistical parameterization. The problem is that the parameters in all of these are fixed, and derived for very specific cloud fields in a multivariate fitting manner. What we do instead is use the spatial context to drive the 3D correction in a flexible and generalizable manner. To be more specific, for the simplest parameterization (fitting a linear regression line to the IPA results from Figure 2), the problem is that the slope varies from scene to scene. You can see this in Figure 8, where the slope of the IPA retrieval is plotted as a function of cloud fraction and other scene parameters. In other words, a single slope parameter does not allow the correction of IPA retrievals. The CNN technique can be regarded as a more complex form of "fitting", where a parameterized correction of 3D effects in COT retrievals is done, in part, as a function of the structure of the cloud field. Still, we appreciate this comment as it made us think about the distinction between simple linear parametrizations and more complex CNN approaches. Once we hear back from the complete review committee, we will add a statement in the revised manuscript. Here is a draft of what that might look like:

It is worth noting that the IPA in Fig. 2 does appear to have a linear relationship with the (retrieved - true) COT which would imply that it is indeed possible to parameterize this effect as a 3D correction. Furthermore, as we discuss in Sect. 1.2, there have been approaches that have attempted to do so, including Iwabuchi and Hayasaka (2002) who introduced a more complex statistical parameterization. However, the underlying problem with such a method is that the parameters are fixed, and derived for very specific cloud fields using multivariate fitting. By contrast, with our proposed CNN (and future iterations of it), the intention is to utilize the existing spatial context in cloud imagery to learn the underlying features that can then be generalized and applied to correct 3D radiative/net horizontal photon transport effects.

Line 207, what are the three aerosol number concentrations?
-> Per the Yamaguchi et al. (2009) paper that conducted the study over the Sulu Sea (and cited in this paragraph), the three aerosol concentrations were 35, 150 and 230 $mg^{-1}$

Line 214-217, why do the authors only use two daytime periods?
-> The Lagrangian LES conducted by Kazil et al. (2021) focus on a cloud state transition (closed- to open-cell stratocumulus) from the first to the second daytime period. The simulations capture the cloud state transition in its entirety, which is sufficient for the work's objectives. The cloud deck dissipates shortly after the end of the simulations (as seen in satellite imagery) and longer simulations would not provide additional data. Finally, the LES in Kazil et al. (2021) use very large domains and sectional (bin) cloud microphysics, which makes them expensive.

Equation(5), In my understanding, this equation calculates the water vapor amount instead of liquid water content. Could the authors help explain?
-> Thank you very much for pointing out this error, we missed this. The equation is supposed to use $mr_{cloud}$ rather than $mr_{water}$. We will make this change in the revised manuscript.

**Response to RC2**

**Dear Reviewer,**

**Thank you for your detailed comments and suggestions for our manuscript. We believe that it was constructive and enabled us to improve the clarity and accessibility of the manuscript. Please find in this document a point-by-point response to your comments. We have detailed our plan to change the manuscript per your suggestions and would like to take this opportunity to once again thank you for your time. Our responses are in bold typeface and black font color while the proposed changes we intend to make in the manuscript are in italics and blue font color.**

**Please note that sections, section numbers, and line numbers referenced in this document are from the preprint version of the manuscript.**

Major concerns:

As a reader/reviewer with very little knowledge about ML, I approached this paper with a strong desire to learn more. And about how ML can be applied to an endeavor that I care about. The fact that none of the authors come with an affiliation in computer science made my expectation even greater. I was, however, disappointed. The key Section 3 was not easy to read. I came out of it feeling that there was either too much or not enough detail. In particular, what I guess is ML jargon was often not explained.

I strongly recommend that the authors make that Section 3 into an Appendix with improvements suggested below (mostly more details), and leave in the main text (thinking of it as "mandatory reading") a well-crafted high-level summary. That summary of what's going on in the "black box" should be just enough to leap into the interesting results presented in Section 4.

**Thank you for your candid feedback. After reviewing the section again, we agree with you, in particular with your view of the machine learning section (Section 3).**

We have been discussing the scope of this section for a while as well. We actually do want to improve the ML section such that it can be understood by a non-Computer-Science reader. For this reason, we do think that much of the content should remain in the body of the paper, whereas details that detract from the flow should be put into the appendix. We will certainly make sure to introduce our terms better so as to avoid jargon and make the content truly accessible. As we state in the manuscript, this work is intended to open up a new approach to correcting 3D effects using machine learning and in particular, Convolutional Neural Networks. Therefore, as the first in a series of works, we believe this paper must contain some details of the machine learning specifics in the main text. Furthermore, for the same reason, we cannot gloss over the machine learning section directly to the results section because that defeats the purpose of this paper.

However, we will take your advice on board and make some changes to that section. For instance, we will make the text more intuitive and organic, rather than directly talking about specifics. We will move those particulars like filter sizes, learning rates, etc. to the appendix. We will now add a new subsection 3.1 for introducing some relevant terminology and explaining why and how CNNs work to solve the problem of biased retrievals. This should ensure that a reader with little to no background knowledge of machine learning should be able to understand the overall structure and its application efficacy to the problem we are trying to solve. At the same time, someone from a computer science background can reproduce or reconstruct the model using the details mentioned in the appendix. We will also update the architecture specifics so they can move into the appendix. Subsection 3.5 titled "Training" will now contain much lesser specifics as a result. The following is a draft of parts of section 3 that has not yet been completely separated into the main text and appendix which will be done once the review period ends:

*3 Architecture & Methodology*

*The Convolutional Neural Network (CNN) is responsible for learning features and patterns that can fit the non-linear relationship between the radiance and the cloud optical thickness. In our case, this is done via various multi-channel convolution layers and non-linear transformations. When a radiance image is fed to the model, it gets passed through these layers, undergoing transformations and changes to size and dimension, until after the final layer when it is compared with the ground truth COT to compute the cost or error. This section will detail the workings of the CNN including its*

*setup and inner workings. But first, we will discuss some of the nomenclatures that will be used throughout the rest of the paper.*

*3.1 Machine Learning Terminology*

*CNNs, at their core, are feature extractors. The goal is to learn the underlying low-dimensional and high-dimensional spatial features in the radiance imagery that when optimized, result in a very close approximation of the COT. In order to extract these features, CNNs employ convolution. Each convolution operation works by moving a sliding window or "kernel" over the input to produce a convolved output or "feature map". Every time the kernel is varied, the features it extracts also vary. A kernel is simply a 2D matrix that stores the coefficients or "weights" to be convolved with the input. To put this mathematically, let us say the weight coefficients in the 2D kernel (the sliding window which moves over the input to perform convolution) of a filter (stacked kernels) are given by $w$. If, say, the kernel size is $K$ x $K$ (meaning $w$ is a $K$ x $K$ matrix), and it is convolving over an $M$ x $M$ input image $x$, then the convolution operation can be written as*

$$z_{u,v} = w * x_{u,v} + b = \sum_{i=0}^{K-1}\sum_{j=0}^{K-1} w_{i,j}\, x_{u-i,\, v-j} + b \qquad (7)$$

*where $b$ is a bias vector. "Biases" are constants (or constant vectors) that are used to offset the output of the convolution. They help reduce the variance and provide flexibility to the network. Convolution computes the dot product over each pixel of the input over a $K$ x $K$ window, offset by a bias value to obtain a single value of the 2D feature map matrix, represented here as $z_{u,v}$. To obtain the next feature map value, the kernel slides over and repeats the operation in Eq. 7. In doing so, the convolution builds and fills out the feature map. Once this feature map is obtained, an activation function $f$ that is typically non-linear is applied to help decide which features of the feature map should be "activated". It also introduces the non-linearity component to convolution. The resulting output is termed the "activation map". Following Eq. 7, this activation is given as:*

$$y = f(z) = f(w * x + b) \qquad (8)$$

*Common activation functions include the sigmoid and $tanh$ functions. For our proposed CNN, we use a type of activation function called ReLU or Rectified Linear Output. It is a function that only activates when the features are non-zero. It can be written as:*

$$f(z) = max(0, z) \tag{9}$$

The features that a filter extracts could be as simple as a horizontal or vertical gradient, or more complex and high-dimensional. When a number of kernels are stacked channel-wise, they are called "filters". The major advantage of using a CNN is that we do not need to manually set the values of these kernels or filters beforehand; it learns these values through optimization over time. This period during which the CNN learns how to best set the weights that will result in the lowest error in its estimation of COT is termed "training". More background information on the training process is provided in Appendix B.

**3.2 Architecture**

Our CNN can be explained in two aspects: (1) the architecture and (2) the training. The architecture is derived from an existing U-Net design (Ronneberger et al., 2015). Figure 4 shows an illustration of the architecture. We opted for a U-Net style architecture for three main reasons: 1) the model complexity is lower than other architectures which thereby increases computational speed during both training and evaluation (inference); 2) the use of concatenation layers linking features learned by different stages helps the model learn new features having more information without increasing layer depth; and 3) the U-Net has been proven to be a state-of-the-art model for segmentation problems, especially in the medical field Litjens et al. (2017).

**3.2.1 Contracting Path**

The U-Net architecture in Fig. 4 can be broadly thought of as two distinct halves in the U-shape: a contracting branch on the left that can be viewed as an "encoder", and an expanding branch on the right that can be viewed as a "decoder". The encoder, i.e the left contracting half, progressively reduces or "contracts" the spatial dimensions while increasing the feature dimensions. The decoder, on the other hand, does the opposite. This is because the objective of the encoder is to translate the features of the radiance imagery into a low-dimensional representation (at the bottom of the U-shape). This representation is the result of learning features of cloudy and non-cloudy regions at multiple scales. The decoder then projects these low resolution features back to the pixel space so as to classify each pixel into a COT bin.

*The contracting path (the encoder) is composed of a series of convolutional blocks separated by pooling layers. Each convolutional block has two sequential identical sets of a 2D convolution layer, batch normalization layer and an activation layer in that order (Convolution -> Normalization -> Activation -> Convolution -> Normalization -> Activation). Details about each individual layer in the encoder are provided in Appendix C1.*

*3.2.2 Expanding Path*

*The expanding path (the decoder) is the right half of the architecture, composed of a series of decoding stages and the same convolutional blocks from the contracting path. Each decoder stage enlarges or upsamples the spatial dimensions by a factor of 2 using bilinear interpolation. For instance, after the end of the contracting path at the bottom of the U-shape, the dimensions of the feature map are 4 x 4 x 1024. After we upsample, the new dimensions become 8 x 8 x 1024. In addition, an operation called "transposed convolution" is performed after upsampling to provide a learnable set of parameters to the upsampling process as interpolation is not intrinsically learnable. Transposed convolution is further explained in Appendix A. The transposed convolution step halves the number of feature maps, but we reinforce them with the corresponding feature maps of the same spatial dimensions from the encoding path on the left using concatenation (depicted as grey arrows in Fig. 4). This concatenation operation between layers (often referred to as "skip connections" in machine learning literature) helps add extra information to the upsampling stage from the encoder side. We then pass the concatenated feature maps through a convolution block, just like we did on the encoder side. We repeat this upsampling, concatenation, and convolution process until we reach the desired spatial size of 64 x 64. Once the original resolution is reached, a final convolution layer with 36 filters is applied, resulting in an output of 64 x 64 x 36. Since this is a segmentation approach, we need to translate this output to probabilities where each pixel has a probability distribution across the 36 COT classes. In other words, the network needs to tell us how likely a pixel is to belong to a range of COT bins. We accomplish this probability translation by applying a softmax function to the output. This function can be written as:*

$$f(z_i) = \frac{e^{z_i}}{\sum_{j=0}^{N-1} e^{z_j}}$$

[revised manuscript text omitted]

*Appendix C2: Decoder (Expanding Path)*

*The upsampling operation is done by bilinear interpolation in the decoder. The spatial dimensions get doubled each time this operation is performed. Immediately after interpolation, we apply a transposed convolution with a 2 x 2 kernel and a 1 x 1 stride and also pad the operation. Again, this is activated by the ReLU function after batch normalization. Next, the corresponding feature maps from the encoder are concatenated and the same configuration of the convolution block from the encoder - - 3 x 3 kernel with single stride and ReLU activation - is used to learn more features from the concatenated output.*

Sequential comments:

* Fig. 1: The radiance scale is missing (best to use "BFR" units, pi I / mu_0 F_0).

**Thank you for pointing this out, we will add a color bar (visually accessible to color-deficient persons) as well as units to both radiance and COT. Here is what that will look like:**

[Figure]

* Fig. 1: To better show the IPA underestimation, maybe add a 4th panel: same as (c) but with a stretched scale.

**We do show the under-estimation in Fig. 2 using a scatter plot which quantizes the scale of under-estimation better than a visual comparison. That said, we will consider modifying the figure to have a better-scaled panel.**

* Fig. 1 and elsewhere: Avoid the "rainbow" color scale that does not work well in B&W print, nor for color-blind persons. Hint: the default "green-yellow" scale in python avoids these pitfalls.

**Thank you for pointing this out, we will update the figures in the revised manuscript to be more accessible to color-blind persons. An example of the modified color scheme/scale is shown above (Figure 1).**

* Section 1.2: The history of efforts to mitigate 3D RT effects is interesting to read. Someday it would be nice to have a more exhaustive version, but here at least one approach antedates BL95 and is worth mentioning, namely:

Cahalan, R.F., 1994. Bounded cascade clouds: Albedo and effective thickness. Nonlinear Processes in Geophysics, 1(2/3), pp. 156-167.

Interestingly, Cahalan's solution involves a multiplicative prefactor, as in (4), rather than BL95's scaling exponent.

**Yes, we agree that this is a relevant paper that we should cite. We will make this change in the revised manuscript as follows in section 1.2.**

*Cahalan (1994) analyzed marine stratocumulus clouds and arrived at a parameterization that utilizes the ratio of mean cloud optical thickness to the cloud effective thickness. This ratio, χ, can be characterized analytically by the cloud fractal parameters, and parametrically by the standard deviation of the logarithmic cloud optical thickness.*

\* Section 1.3: This too is an interesting read that contrasts (physics-based) cloud tomography and (statistics-based) neural nets.  The former is a pretty recent development with the real breakthrough paper being:

Levis, A., Schechner, Y.Y., Aides, A. and Davis, A.B., 2015. Airborne three-dimensional cloud tomography. In Proceedings of the IEEE International Conference on Computer Vision (pp. 3379-3387).

A paper of special interest here that uses a CNN rather than 3D RT in the cloud tomography per se is:

Sde-Chen, Y., Schechner, Y.Y., Holodovsky, V. and Eytan, E., 2021. 3DeepCT: Learning volumetric scattering tomography of clouds. In Proceedings of the IEEE/CVF International Conference on Computer Vision (pp. 5671-5682).

**This is certainly a very interesting and relevant paper that lies at the intersection of tomography and CNNs. We will cite their work in the revised manuscript as follows in section 1.3:**

*Sde-Chen et al. (2021) combined the two worlds of CNNs and tomography to reconstruct the 3D cloud extinction field using multi-view satellite images.*

\* Fig. 2: What happens to the IPA when true COT > 40? We need to see that for a fair visual comparison with CNN.

The IPA continues to under-estimate the COT when COT > 40 as well, and in fact due to the nature of the retrieval, it becomes worse for higher COT. We clipped the higher COTs to maintain the figure aspect ratio and because we felt the point that we were trying to establish about the scale of the IPA under-estimation was made. For a fair comparison, we will clip the true COT to a number suitable for both retrievals, as shown below, and update it accordingly in the revised manuscript:

[Figure]

* Fig. 2: How well does the prediction in (4) for the slope work here, in comparison with the empirical IPA vs true COT slope?

We show the CNN's retrieval from section 4.3 in Fig. 2 as we consider this to be the "best" model i.e the least error-prone and generalizable/robust to different aspect ratios. We show the slope analysis in section 4.3 and NOT in the introduction as we have not yet defined the slope term. Furthermore, since the intended takeaway from this plot in Fig. 2 is that the IPA under-estimates the COT, we did not feel the need to include a slope plot here. We will note that we do introduce the slope term in Section 3.3. Very briefly, we use the term "slope" to mean the slope obtained when a linear regression line is fitted against the true

**COT vs (retrieved - true) COT. An ideal slope would be 0, indicated by the dashed black line in Fig. 2. As we mention in Section 3.3, "for the case shown in Fig. 2, for the IPA retrieval (green scatter), the linear regression slope (with the true COT subtracted from the retrieved COT) "a" is - 0.79". In contrast, the CNN retrieval has a slope of -0.03. This quantization clearly shows the degree of improvement that the CNN is capable of achieving.**

* Fig. 2: To the eye, it looks like, although biased low, the dispersion around the IPA retrieval is much smaller than around the (unbiased) CNN retrieval. Why? This looks like an opportunity to get the best of both approaches.

**Yes, this is a good observation. In this paper, we are not attempting to optimize the CNN for minimal dispersion as we are not after an accurate pixel-level retrieval. Rather, we want to propose a model that is capable of reducing or mitigating the bias while estimating the COTs for "mini-domains". One could introduce a direct error term such as root mean squared error (RMSE) or mean absolute error (MAE) in the cost function if the goal is to reduce the scatter.**

* lines 190-193: I couldn't comprehend the interchangeable use of "level of coarsening" and "aspect ratio" (AR) until I downloaded and browsed BL95. As I understand, BL95 is based on cloud models generated from 2D (Landsat) imagery. So, there is a user choice of how much geometric thickness (h) to assign to the clouds. In that case, talking about AR makes sense, and BL95 modulated it by varying h. But here the LES-generated clouds are inherently 3D. So, talking about coarse-graining (in the horizontal plane) makes sense. But the AR is something different unless all that is taken from the LES is the 2D COT field and cloud thickness in the 3rd dimension is assigned and held constant, like in BL95. If that is the case, I missed it.

**Yes, we now see how this could be confusing. As you point out, the user gets to choose the cloud geometric thickness (h) when using Landsat imagery as it is 2D. The BL95 paper demonstrates this very well and one can see how the aspect ratio changes when h changes. By contrast, LES produces 3D clouds. We change the aspect ratio (cloud height divided by width) by \*only\* coarsening the width or the horizontal resolution. As we mention in lines 186 and 187, when we coarsen the horizontal resolution by a factor of 2, the aspect ratio gets halved. When we take one step further and coarsen by an additional factor of 2 (and therefore a total coarsening factor of 4), the aspect ratio gets reduced by a factor of 4 from the original. We do not change the vertical resolution which is kept constant. Therefore, the resulting data has 3 pools corresponding to the native resolution and two levels of coarsening factors. We will add a short explanation how aspect**

**ratio and level of coarsening relate to each other, as follows in section 2.1, line 185. Here is a draft of what the new text would look like:**

*For CNN training, the original number of samples is very low. Therefore, we augmented the native-resolution training pairs by horizontally coarsening the fields by a factor of 2, such that each original 100 m x 100 m cell was assigned a spatial extent of 400 m x 400 m and then split into four cells, leaving the vertical resolution of the fields (40 m) intact. In addition to providing additional training pairs after sub-sampling as described for the native-resolution data, this coarsening procedure also effectively generates horizontally smoother cloud fields while halving the cloud aspect ratio (cloud height divided by cloud width) since we only change the horizontal resolution. In other words, one of the key drivers for 3D COT biases as described by BL95 and others is systematically changed in the training data to introduce some training data diversity. A subsequent second coarsening step introduces another level of coarsening and the aspect ratio has now reduced by a factor of 4 from the original. The three data sets, labeled 1 x 1 (native resolution), 2 x 2, and 4 x 4, respectively, are used separately (Sect. 4.1) to examine the impact of the cloud aspect ratio on the retrieval performance, and together (Sect. 4.2) to assess the impact of training sample number along with algorithm robustness and accuracy for a physically more diverse data set. A more consolidated version of the three data sets is evaluated to decrease training time (Sect. 4.3).*

\* line 208: Not an expert here, but I thought that LES microphysics schemes were either "bulk" or "bin" and, in the _former_ case, they can be either 1- or 2-moment. Please clarify "two-moment _bin_ microphysics".

**The bulk scheme can be either 1 or 2 moment (or n-th moments; e.g., 3-moment scheme exists). The bin scheme can be two-moment as well. In a two-moment bins scheme, the size distribution of both mass mixing ratio and number concentration is represented by bins. One-moment bin scheme uses a bin representation of mass mixing ratio and diagnose number. A detailed description of the two-moment bin scheme is given in page 12,246 of Yamaguchi et al. (2019) that is cited in that paragraph.**

* Section 2.1.2: Can I suggest one figure here to visualize the important differences with the Sulu Sea LES clouds? Something like Fig. 3 for the Sulu Sea simulations.

**Yes, this is a great suggestion. We will add a new figure to illustrate a sample dataset from the Atlantic. It should look similar to this (color maps are compliant with accessibility):**

[Figure]

**Figure #. a-d: Lagrangian LES COT fields (600 nm) from the Atlantic taken during the CLARIFY campaign simulating closed-cell stratocumulus cloud deck transitioning to a broken open-cell cloud deck; e: Synthetic radiance calculations (Red = 600 nm) with 3D, shown for the 6.4 km x 6.4 km sub-domain (white box) in (c).**

* Eq. (5): I think you mean r_cloud, not r_water, and maybe "." like in (6), not "*".

**Thank you for noting this, it is indeed a mistake that will be corrected in the revised manuscript.**

* Eqs. (5-6): Why not use the more common "q_lw" and "q_wv" for your mixing ratios? And, accordingly, the usual "Q_ext" for the Mie efficiency factor?

**Thank you for this feedback, we will update the notations as follows in the revised manuscript:**

**Cloud water vapor mixing ratio:** $q_v$

**Cloud liquid water mixing ratio:** $q_l$

**Mie efficiency factor:** $Q_{ext}$

* line 286: Up to 9 km? Is it 6.4 x √2? If so, say it.

**Yes, it is indeed 6.4 x √2. We will address this comment when we rewrite the section to make it friendlier to a non-machine-learning audience. A draft of the rewrite of section 3 is provided above in response to the first question.**

* Fig. 4: Great start for understanding the ML technique used here! However, still too many questions and undefined concepts for the non-cognoscente:

How do you get the 64 layers from a single one in step #1? Can it be another number? (I understand the subsequent doubling and halving.)

What is ReLU Activation?

What is Batch Normalization?

(Maybe better to have different colors for these two operations?)

**Thank you for being specific about the problematic items in this section, this is very helpful. As noted earlier, we will be splitting the machine learning section between the main text and appendix as suggested. The main text will contain more of an overview, whereas details about the ReLU activation, convolutions and normalization will be added to the appendix.**

\* Section 3.2: "cross-entropy" is explained, but not "one-hot encoding", nor is "softmax activation".

**We actually do explain the one-hot encoding approach and the softmax activation in sections 3.4 and 3.6 respectively. We will add a note in section 3.2 to refer to these sections for details about those techniques. Additionally, we will add more details about the softmax activation function during a rewrite of the ML section (Sect. 3). A draft of the rewrite is available at the beginning of this document.**

\* Eq. (11): Does alpha depend on i or c? If so, which and how? (And add the appropriate subscript.) If not, it can be factored out.

**The alpha term is used as a weighting factor that is dependent on the true COT classes. This is because focal loss was specifically developed by Lin et al. (2017) to overcome class imbalance which is relevant for our application as we have an overwhelming majority of non-cloudy pixels. We will add the appropriate subscript and text to make this clear. This is what we expect the new additional text and equations to look like at line 336:**

*An $\alpha_t$ term is added as an additional weighting factor. We make the weighting factor dependent on the true binary COT pixel probability $p_{i,c}$:*

$$\alpha_t = \alpha \cdot p_{i,c} + (1 - \alpha) \cdot (1 - p_{i,c}) \qquad (11)$$

*and we set α to 0.25 as recommended by Lin et al. (2017). The final version of our loss function can be written as:*

$$FL = -\sum_{i \in I}\sum_{c \in C} \alpha_t (1 - \hat{p}_{i,c})^\gamma p_{i,c} \log(\hat{p}_{i,c}) \qquad (12)$$

\* Fig. 5c: What is the top layer? Looks like a binary cloud mask resulting from all the COT classes.

**The top layer would be the pixels falling between 0 < COT ≤ 0.1 . In other words, this would be considered the "background class" of "non-cloudy" pixels. We will add a note in that section to help the reader understand this figure better.**

* line 403: Delete "resolution" (it is the domain size).

**We will make this change in the updated manuscript.**

* Below Fig. 6: Please tell us a little about the "Adams" optimizer.

**We took your suggestion and decided that It is better to explain details about the optimizer in the appendix as it is not necessary to understand this optimizer for the purposes of this application. The following text will be included in the appendix and referenced in the main text at line 418:**

*The Adam optimizer is an extension of the commonly used gradient descent algorithm that is used to train machine learning algorithms. In a regular stochastic gradient descent algorithm, a single constant learning rate α is used across all the weights in the model throughout training.*

*The Adam optimizer, which stands for Adaptive Moment Estimation, uses a slightly different approach when it comes to gradients. It calculates a moving average of the mean (the first moment) of the gradients, and a moving average of the squared gradients (the second moment). It uses two adjustable parameters $\beta_1$ and $\beta_2$ to control the rate at which these averages. From the Kingma and Ba (2015) paper, we can glean that this algorithm has numerous advantages, most notably that it has no memory requirements and is appropriate for noisy gradients. Over time, this optimization algorithm has come to be popular and is commonly used as the default optimization for numerous ML problems.*

* Above Eq. (16): One COT bin (#27) is finally given explicitly. What about the others? Are they linearly sampled? Logarithmically? Surely this discretization of the COT scale also has to be somehow optimized.

**This is a great observation that we have been exploring as well. The exact bin values and classes we used were as follows:**

| Range of COT | Binned Class |
|---|---|
| [0.0, 0.1) | 0 |
| [0.1, 0.2) | 1 |
| [0.2, 0.3) | 2 |
| [0.3, 0.4) | 3 |
| [0.4, 0.5) | 4 |
| [0.5, 0.6) | 5 |
| [0.6, 0.7) | 6 |
| [0.7, 0.8) | 7 |
| [0.8, 0.9) | 8 |
| [0.9, 1.0) | 9 |
| [1.0, 2.0) | 10 |
| [2.0, 3.0) | 11 |
| [3.0, 4.0) | 12 |
| [4.0, 5.0) | 13 |
| [5.0, 6.0) | 14 |
| [6.0, 7.0) | 15 |
| [7.0, 8.0) | 16 |
| [8.0, 9.0) | 17 |
| [9.0, 10.0) | 18 |
| [10.0, 12.0) | 19 |
| [12.0, 14.0) | 20 |
| [14.0, 16.0) | 21 |
| [16.0, 18.0) | 22 |
| [18.0, 20.0) | 23 |

| | |
|---|---|
| [20.0, 25.0) | 24 |
| [25.0, 30.0) | 25 |
| [30.0, 35.0) | 26 |
| [35.0, 40.0) | 27 |
| [40.0, 45,0) | 28 |
| [45.0, 50.0) | 29 |
| [50.0, 60.0) | 30 |
| [60.0, 70.0) | 31 |
| [70.0, 80.0) | 32 |
| [80.0, 90.0) | 33 |
| [90.0, 100) | 34 |
| [100, 200) | 35 |

**It is true that it is possible to further optimize the binning mechanism. Especially considering the impact it has during the class-to-COT translation after the model is trained. As this paper is meant to show the feasibility of the CNN to the problem of correcting biased retrievals, we do not explore this particular aspect in detail. The bins themselves would be dependent on the range of COT data available and therefore case-dependent. This makes it harder to have a predefined or "optimized" set of binning criteria but it is possible to create a generalization of a set of bins.**

\* Section 4.1, and below: Better to use "coarsening factor" than "aspect ratio" (see comment above for lines 190-193).

**We do agree that the terms "coarsening factor" and "aspect ratio" are used interchangeably in places where it might not be appropriate, especially when prefixed with the 2x2 or 4x4 terms. We will rectify this ambiguity in the revised manuscript.**

* line 447 and Figs. ≥7: Cloud Variability is an interesting non-dimensional quantity. It seems to have an upper bound of 2, but that isn't clear from the definition. Please clarify.

**Cloud variability is useful as it allows us to evaluate the performance of the model on a metric that relies on the inhomogeneity of the data. As stated in the same paragraph, we compute cloud variability as the ratio of the standard deviation of cloudy pixels (COT > 0.1) to the cloud fraction. There is no upper bound on cloud variability. In Fig. 12, we can see that it is in fact clipped to just over 6, while other experiments using different subsets of the Sulu Sea have different upper limits. The clipping is only done on the upper bound for illustration where there is an insignificant number of samples past that bound. We do mention these in the manuscript.**

* Section 4.2: Why is the number of scenes used for training described as "cloud morphology"? (That term does come up later on, in p. 25 in Sect. 4.4 where clouds from different regions are contrasted.)

**We understand the confusion that might arise here with the two terms. The number of scenes is not the same as cloud morphology. Each of the six 480x480 scenes has a different cloud morphology as explained in section 2.1. And each scene has a different set of values for wind shear, cloud fraction and aerosol concentrations. The intention behind section 4.2 is to explore how important cloud morphology is to learning the underlying features. This is why we keep the aspect ratio constant and only vary the number of diverse cloud morphologies the model is exposed to. One worry with machine learning algorithms is that they can "overfit". This is when the model appears to memorize the training data and does not learn any of the higher dimensional features needed to generalize to a dataset outside the training envelope and is therefore not robust to changes. By exploring the 1 cloud morphology variation vs 5 cloud morphology variations setup, we are able to definitely say if the model is capable of generalizing to new datasets such as the Atlantic.**

* Fig. 9, caption: What are the red dots?

**Each red dot represents a 64x64 radiance image. We will mention this in the paper as follows in the captions for the figure:**

*Figure 9. A distribution of standard deviation of reflectance ($Ref_{std}$) vs mean of reflectance ($Ref_{mean}$) for the gridded and sampled data set. Each red scatter dot represents a 64 x 64 reflectance image belonging to the consolidated data set consisting of the 1 x 1, 2 x 2 and 4 x 4 coarsening factor data pools.*

\* line 576: typo in "erroneous"

**We will fix this typo in the revised manuscript.**

\* line 605: Maybe the contradiction found here with BL95 has to do with the key difference (discussed previously) between their use of "aspect ratio" and the "coarsening level" that it is equated to here?

**As discussed earlier, the BL95 paper changes the aspect ratio by changing the cloud geometric thickness and we do so by introducing coarsening for the horizontal resolution. But, perhaps another notable point of difference is that we use 3D fields from LES while BL95 uses 2D Landsat imagery. There is also the significant factor that the Landsat imagery used in BL95 was mostly stratocumulus whereas we deal with isolated cumulus, which is arguably more complex. In our study, we found that the IPA (1D) retrieval remained consistently worse and did not necessarily get worse with changes in aspect ratio. However, due to the many points of difference between our paper and BL95, as you point out, we cannot definitively say that this discrepancy can be directly compared against each other. We will make a note of this in the conclusion section. Here is what a draft of that updated text will look like (starting from line 604):**

*From this experiment, we found that changing aspect ratios did not significantly alter the physics to the detriment of retrieval fidelity, despite the findings of BL95. However, we must note that there are significant differences between the BL95 study and ours. Most notably, BL95 used 2D Landsat imagery with mostly stratocumulus cloud fields while we use 3D LES with isolated cumulus clouds. It should also be noted that the BL95 paper varies cloud geometric thickness to change the aspect ratio while we vary the horizontal resolution keeping other dimensions constant. Therefore, we cannot conclusively say that aspect ratio was the sole direct cause of this discrepancy in the IPA retrieval performance.*